# N:P STOICHIOMETRY AND HABITAT EFFECTS ON MEDITERRANEAN SAVANNA SEASONAL ROOT DYNAMICS

[1] RICHARD K F NAIR, [1] KENDALYNN A MORRIS, [1] MARTIN HERTEL, [1] YUNPENG LUO, [3] GERARDO MORENO, [1] MARKUS REICHSTEIN, [1,2] MARION SCHRUMPF, and [1] MIRCO MIGLIAVACCA

[1] Max Planck Institute for Biogeochemistry, Department Biogeochemical Integration 07745 Jena, Germany
[2] Max Planck Institute for Biogeochemistry, Department Biogeochemical Processes 07745 Jena, Germany
[3] Forest Research Group – INDEHESA University of Extremadura, Plasencia, Spain
Correspondence: RICHARD NAIR (rnair@bgc-jena.mpg.de)

**Abstract.**

Mediterranean grasslands are highly seasonal and co-limited by water and nutrients. In such systems, little is known about root dynamics which may depend on individual plant properties and environment as well as seasonal water shortages and site fertility. Patterns of root biomass and activity are affected by the presence of scattered trees, grazing, site management, and chronic nitrogen deposition, all of which can affect nutrient ratios and potentially cause development of nitrogen:phosphorus (N:P) imbalances in ecosystem stoichiometry.

In this study we combined observations from minirhizotrons with root measurements from direct soil cores and ingrowth cores, along with measures of above-ground biomass to investigate seasonal root dynamics and root:shoot ratios in a Mediterranean tree-grass 'savanna'. We investigated responses to soil fertility, using nutrient manipulation (N / NP addition) and spatial microhabitat treatments between open pasture and under tree canopy microhabitats. Root dynamics over time were also compared with indices of above-ground growth drawn from proximal remote sensing.

Results show distinct differences in root dynamics and biomass between treatments and microhabitats. Root biomass was higher with N additions, but did not differ from the control with NP additions in early spring. By the end of the growing season root biomass had increased with NP in open pastures but not higher than N added alone. In contrast, root length density (RLD) in pastures responded stronger to the NP than N only addition, while beneath trees root biomass tended to be higher with only N. Even though root biomass increased, root:shoot ratio decreased under nutrient treatments. Timing of root and shoot growth was reasonably well paired, although in autumn root growth appeared to be substantially slower than 'regreening' of the system. We interpret these differences as a shift in community structure and/or root traits under changing stoichiometry induced by the fertilization. We also consider seasonal (phenology) differences in the strength and direction of effects observed.

## 1 INTRODUCTION

Terrestrial semi-arid ecosystems are important determinants of the interannual variability in the land carbon (C) sink (Ahlstrom et al., 2015) as variation in C uptake is controlled by plant phenology (Randerson et al., 1997; Richardson et al., 2013) which

responds to seasonal and interannual variation in climate conditions. In Iberia, one example of these semi-arid ecosystems is a managed agro-silvopastoral savanna, known as 'dehesa' or 'montado' (Spanish and Portugese names, respectively). Similar systems are common in other mediterranean countries (den Herder et al., 2017) and worldwide (Campos et al., 2013; Porqueddu et al., 2016).

Dehesas consist of 20-40 % *Quercus ilex*. Ballota (Desf.) and *Quercus suber* (L.) canopy cover with seasonally variable intercanopy grassland maintained by livestock grazing (Moreno and Pulido, 2009). They are a man-made conversion from oak forest (Joffre et al., 1999) as a combination of low seasonal water availability and typically low nutrient availability limits conversion to other cover types (Eagleson and Segarra, 1985). Their complex structure and multiple seasonally limiting resources (water in summer, nutrients in wetter periods of the year) means these regions are also particularly badly represented in vegetation components of predictive models (Beringer et al., 2011). These areas around the Mediterranean basin are also especially vulnerable to climate change (Giorgi and Lionello, 2008; Sillmann et al., 2013) due to increasing aridity, increases in temperature (Peñuelas et al., 2018) and other environmental changes due to human activity. Particular among these are nitrogen:phosphorus (N:P) imbalances (Peñuelas et al., 2013), which result from chronic N inputs (from deposition and management) at a higher rate than P inputs. These stoichiometric imbalances may have major impacts on plant functioning, but must be considered within the context of both major structural, micrometorological and soil fertility-associated differences between tree- and grass- microsites (Moreno et al., 2013) and the impact of severe summer droughts.

Plant phenology studies tend to focus on above-ground organs (Radville et al., 2016), despite the fact that below-ground systems are the main source of carbon dioxide emissions to the atmosphere (Schlesinger and Andrews, 2000), contain 2/3 of the world's C stocks (Batjes, 1996) and are the site of plant uptake of both water and nutrients via roots. In grasslands (and by extension, grass-dominated systems such as savannas), below-ground systems are also the site of most competition between individuals (Mokany et al., 2006), the major short-term sink for recently fixed C due to high ratios of roots to shoots (Hui and Jackson, 2006; Mokany et al., 2006), the main source of litter (Casals et al., 2010), and the main contributor to long-term soil C stocks (Rasse and Smucker, 1998), and a major site of niche differentiation between plant forms (Moreno et al., 2013). In most ecosystems, root biomass changes substantially throughout the year, although understanding drivers of this phenology is limited, especially when using quantitative metrics (Radville et al., 2016). In many cases, root growth is desynchronized from production of shoots (Blume-Werry et al., 2017; Mccormack et al., 2017; Steinaker and Wilson, 2008) and linkages between root function and root dynamics are often poorly understood. As a major function of roots is nutrient uptake, supplying resources which are often limiting, nutrient availability may play an important role in regulating the timing and magnitude of root production. In seasonally arid and Mediterranean systems, plants are thought to be co-limited by N and P (Ries and Shugart, 2008; Sardans et al., 2012; Sardans and Peñuelas, 2013) but there is considerably less information in root responses than commonly-measured above-ground parameters. Roots may respond in different ways to shoots, particularly under drought (Gargallo-Garriga et al., 2014), and generally, responses below-ground are less consistent than above-ground. This may relate to the balance of co-limitation of nitrogen (N) and phosphorus (P), and water, which are all acquired by roots, and vary in availability throughout the year (Ries and Shugart, 2008). Hence, while N inputs may result in generally higher biomass in

Mediterranean grasslands (Dukes et al., 2005), it is unknown to what extent these above-ground patterns are reflected in below-ground development.

In Mediterranean regions, grass phenology typically centres around a summer dormancy, with a dry down in late spring and a 'green up' in autumn following the onset of rains. In the more 'interior' locations of Iberia (to some extent true in the experimental site used in this study (Luo et al., 2018)), cool winters lead to an additional temperature-driven winter dormancy (Milla et al., 2010; Thompson, 2005). The main growing season is in spring for oaks and pasture species (Oliveira et al., 1994; Orshan, 1989) before an arid summer with senescence of all the annual components. Roots in such systems are highly spatially segregated (Moreno et al., 2005) between herbaceous plants, which dominate the upper 30 cm of soil, and trees, with deep roots which can access water sources (Castro-Díez et al., 2005) through the dry summer. Due to the relatively low canopy cover, high plant diversity, and common use as grazing pasture, microsites may differ substantially in soil properties, with trees having a strong influence due to large inputs of oak litter and nutrient transport from deeper soil layers (Gallardo, 2003). While this litter is more recalcitrant than grasses, tree microsites also tend to have higher nutrient availability (Gallardo et al., 2000). In general, trees also have less tightly coupled above- and below- ground phenology than grasses (Steinaker and Wilson, 2008; Steinaker et al., 2010) due different abilities to store carbohydrates and nutrients over longer lifespans. A meta analysis (Abramoff and Finzi, 2015) of root and shoot phenology found that in Mediterranean systems (a very coarse definition including both forests and deserts across only 4 studies), peak root growth tended to lag behind peak shoot growth by over a month on average. Overall, shoots were produced in a peak during the main (spring) growing season while root production continued through the year.

There are very few quantitative comparisons of root and shoot phenology in Mediterranean ecosystems, fewer in treatment experiments, and to our knowledge none in mixed tree-grass savanna systems where both microsite factors and extremes of temperature and water availability may promote different responses above- and below- ground. One reason for the relative lack of information on fine root growth patterns in many systems is the difficulty of sampling an opaque, three-dimensional environment usually only accessible from above. Root biomass typically varies spatially due to resource and environmental heterogeneity (Hodge, 2004) and in biologically diverse systems, small scale patchiness is increased by individual species with different root habits. While root biomass is a direct measure of root C stocks, it is also generally highly inconsistent in experimental responses (e.g. Arnone et al., 2000; Mueller et al., 2018). Other physical attributes (such as spatial distribution of root systems, or traits such as root length density (RLD)) may be more relevant for explaining plant function and not just standing biomass as they relate to functional properties; for example as root diameter and density vary, RLD relates more directly than biomass to soil exploration. However, all field methods to measure roots have significant downsides (Mancuso, 2012); biomass methods require destructive sampling as roots must be extracted from the soil and cleaned, while visual methods such as minirhizotrons require both pre-installation of observatories, consideration of artefacts and require lengthy and somewhat subjective post-processing.

We took advantage of a nutrient (+N, +NP) experiment located in a dehesa, where fertilization had been applied on an eddy-covariance footprint scale (described in El-Madany et al., 2018), to study the effect of nutrient additions on root growth and phenology. If a basic economic analogy for plant resource assignment (Bloom et al., 1985) is applicable, resources are allocated to maximise uptake of limiting nutrients, (Fichtner and Schulze, 1992; Shipley and Meziane, 2002). As we expected our system to be limited by N (Dukes et al., 2005), when N is added plants should be more limited by P. When N and P were added together, the system should be similar to its original state in terms of the stoichiometry of these nutrients but with more N and P available. A summary of this 'addition-limitation' hypothesis is presented in Figure 1 (panel a.).

We used a combination of minirhizotrons (providing strict repeatability and information on root phenology), soil 'ingrowth' cores (measuring root production in root-free soil analogous to recolonization around minirhizotrons) and direct soil coring (providing direct biomass measurements). We sampled mostly the herbaceous layer roots, which dominated root biomass in the 0-40 cm depths we studied (Moreno et al., 2005), although these were also likely to be the pool most variable on a seasonal and interannual basis. In similar systems most nutrients (Jackson et al., 1988) and root length density (Moreno et al., 2005) is found in surface soils.

We hypothesized that 1.1) fine root biomass followed an annual cycle, developing through a growing season which begins in autumn, and ends with summer drying the following calendar year and that 1.2) root phenology would be closely synchronized with shoot phenology of the predominantly annual herbaceous layer. For the nutrient additions, we expected that 2.1) nutrient addition of +N and +NP would alter overall root production. If the site was classically co-limited, we would expect to see larger increases in root mass in +NP, than +N. Additionally, 2.2) nutrient addition should decrease overall root:shoot ratios in +NP, maintaining site stoichiometric ratios. With +N, we expected the system to shift into a more P-limited state (otherwise alleviated by the P addition in the +NP treatment, so 2.3) different biomass and RLD effects were expected for +N and +NP additions. These effects were also expected to be 2.4) affected by tree and pasture microsites, as higher expected nutrient availabilities under trees could buffer responses to the treatment additions.

## 2 METHODOLOGY

### 2.1 Study site, and Nutrient Treatments

We worked at Majadas de Tiétar, a long-term experimental site in Extremadura, Spain (central location: 39°56'25.12" N, 5°46'28.70"W) from December 2016 until May 2018. The site is a typical Spanish dehesa with a low density of oak trees (*Quercus ilex (L.)*) at ~20 trees ha$^{-1}$. The herbaceous layer is a seasonally changing mixture of species (Perez-Priego et al., 2015) grazed by cows (< 0.3 cows ha$^{-1}$) during productive seasons which are pastured elsewhere during dry periods. In this study, cows were absent from June 2017 – December 2017. The mean annual temperature is 16°C, mean annual precipitation is ~650 mm, mostly falling between October and April, with a typical Mediterranean climate of long, hot, dry summers and mild, wet winters. Soils are classified as an Abruptic Luvisol with a sandy upper layer (~5% clay, 20% silt, 75% sand between 0-20 cm), and a clay layer between 30 and 60 cm.

The site has been an eddy covariance (EC)-instrumented FLUXNET site ('ES-Lma') since 2004 with two additional 15m towers since 2013 (El-Madany et al., 2018) used for a large-scale ecosystem manipulation experiment. The objective of this site-level experiment was to understand ecosystem responses at the EC-footprint (20 hectare) scale to fertilization treatments designed to disrupt ecosystem nutrient stoichiometry. N additions (as Ca-ammonium nitrate fertilizer), and N and P additions
(as ammonium nitrate and triple superphosphate fertilizer) were added to one EC footprint each in growing seasons 2014/2015 (100 kg N ha$^{-1}$, 50 kg P ha$^{-1}$) and 2015/2016 (20 kg N ha$^{-1}$, 10 kg P ha$^{-1}$) resulting in the three fertilization treatments (El-Madany et al., 2018) used in this study; control, +N and +NP.

The overall site-level experiment was designed to study the impact of stoichiometric N:P imbalance, with the nutrient additions providing: standard conditions (control), high N:P expected to develop P limitation (+N), and relieved P limitation
(+NP). There was no direct P-only addition but previous publications at the site have demonstrated using a small scale fully factorial experiment a lack of an effect of P alone on vegetation structure and soil properties beyond increases in leaf tissue P and P turnover rates (Perez-Priego et al., 2015; Weiner et al., 2018).

The three tower footprints are instrumented with a variety of instruments to measure large-scale ecosystem properties including phenocams (Luo et al., 2018) using standard Phenocam Network methodology (Sonnentag et al., 2012; Richardson
et al., 2018), as well as smaller-scale measurements of plant biomass, and traits, including the root measurements reported in this study.

## 2.2   Root Observatories

We installed root observatories and made root measurements within the site-level footprints. In order to pair these measurements with the site-level design it was necessary to deploy these within the three continuous footprints, each with a different
stoichiometry treatment (Figure 1). Given the large footprint of EC measurements and for logistical reasons of finding large enough areas with homogenous land cover and the high cost of instrumentation, EC measurements are typically not spatially replicated. Hence all root measurements reported are subsamples within the three footprint treatments at the site and such point measurements are replicated in space to cover site-level variation within treatments. Ecosystem properties and the results of pre-treatment measures (including soil C, N and P, leaf level nutrients (Table 1), and EC fluxes (El-Madany et al., 2018),
indicated that these footprints ( 25 ha) were comparable before the start of the fertilization experiment.

This design means that replicates of root analyses per treatment as conducted inform about the variation within each footprint area and are therefore strictly pseudo-replicates. This violation of independence was an unavoidable constraint of the experiment given the very different spatial scales employed at the field site and ambitions for a larger scale synthesis of data across scales. Ecosystem properties and the results of statistical tests of pre-treatment measures indicated that variation was
greater within than between treatment footprints before application of fertilizers (Table 1), giving confidence on the robustness that the findings presented in this study are not an artifact of the spatial variability within the experimental area, and so we interpret differences between the three footprints as differences due to the nutrient treatments applied.

We installed 12 minirhizotron observatories per treatment (36 in total) in May 2016 (size: 1 m x 10 cm (l x d), walls of 3mm thick transparent plexiglass), 8 months before the first measurement was made. Observatories were arranged in sets of four

around randomly selected individual trees (3 per plot), from around 480 trees per 20 ha treated plot (with a few constraints based on spacing of trees, to ensure unambiguous 'grassland' and 'canopy' microhabitats were selected and to preserve spacing between sites). The median distance between tree microhabitats within a treatment is 116 metres and 60 meters minimum. At each tree, we installed two minirhizotrons per tree in 'tree covered' microhabitats (halfway between the stem and canopy edge), and two observatories in 'open pasture', at least three times the canopy radius from the stem and no closer to any other tree. This design resulted in six replicates in a nested design per combination of nutrient treatments (Control, N, NP) and habitats (tree covered, open pasture), shown in Figure 1, panel b.). We detail the statistical implications of this design in the 'statistical analyses' section.

Each set of four observatories was installed on a roughly north-south axis to minimize daily variation in solar heating, and all observatories were parallel to the stem. We placed a small bag of silica gel on a piece of high-friction plastic weave within each observatory in order to reduce condensation on the inside of the tube. Despite a tight fit at installation, at first sampling (December 2016), winter soil swelling had moved some observatories in wetter microsites, which we immediately repaired with additional stabilization equipment, resulting in no further movement in future campaigns. We excluded these observatories from the first sampling, but included them from March 2017 onwards as processed images were no different to other observatories. Similarly, two observatories (one in the +N, one in the +NP treatment) were damaged while in the field during the study. These observatories were carefully removed and replaced with a spare tube which was photographed from the subsequent campaign onwards.

## 2.3 Minirhizotron Method / Sampling Protocol

We used a custom built minirhizotron camera in our observatories modified from other designs (e.g. Amato et al., 2012). This was a visible light camera (FCB-MA 130-FG, JenCam GmBH, Germany) using a 45° mirror to view the outer tube surface with an adjustable handle allowing movement both on the long axis and in defined rotational positions. In the final sampling presented in this paper, this instrument was replaced by a prototype of a new design eliminating the mirror but otherwise producing comparable images. Images were collected using VREO OneView software in .bmp format at 4192 x 3104 pixel resolution and consisted of an image of the mirror surface. The camera was lit with two rows of LED lights on either side of the mirror. We trimmed images to remove image overlap which also removed areas of poor illumination, resulting in a 'window' of observation 5.25 x 4.14 cm per image. We sampled all observatories on seven dates spanning December 2016 to May 2018, corresponding roughly to bimonthly measurements in sequential phenology phases of winter, early spring, late spring, late summer drought, winter, and early spring. Each time, we calibrated the camera against a grid of known dimensions fixed to an unburied observatory and collected a complete rotational profile of images (8 total) every 5 cm along the minirhizotron (complete coverage) with image midpoints 0 to 23 cm vertical depth and three additional depths (28, 38 and 44 cm), resulting in 2880 images per sampling date. In December 2016 only two depths were measured (10 and 20 cm), while in May 2017, September 2017, March 2018 and May 2018, the entire observatory was imaged. We also took a second set of images at all depth positions from the 'control' treatment alone in mid-May 2017, 10 days after the complete survey in early May. This

was a few days after a major rain event, allowing an observation of the short-term response in fine roots to increased water availability.

## 2.4 Image Analysis

We rescaled each trimmed image to a standard 1123 x 1434 pixel resolution. After manually filtering images for quality (re-
moving those with image artefacts or ambiguity in cover), a 10 x 13 grid (i.e. 112 by 110 pixel) was superimposed over each image, producing 130 squares. Each square was marked as either 'roots present' or 'roots absent' following the criteria that a visually unambiguous, apparently living (i.e. with clearly defined form and no obvious structural breaks) root, crossing at least half the square. The proportion of squares with roots per image was calculated (Root Cover Index, RCI). This required 0-2 minutes per image and had significant time advantages over comparative methods which focus on markup of all roots in the
image. We validated this method against a standard methodology using the open source minirhizotron interpretation software Rootfly 2.02 (Birchfield and Wells, 2011) to mark up all roots in the images, using 200 randomly selected images from each of the March 2017, May 2017, September 2017 datasets for calibration.

## 2.5 Direct Soil Measurements

We collected ancillary measurements of root biomass from two non-concurrent methods, sampling within 1 m of the minirhi-
zotron tubes; soil cores (Dec 2016, March 2017) and two rounds of ingrowth cores installed in December 2016 (removed May 2017, December 2017) and December 2017 (removed March 2018, May 2018). These were direct measures of root properties but highly labor intensive and impractical in drought periods. Ingrowth cores were installed by removing a 13 cm, 4.5 cm wide soil column with an auger, homogenizing the soil and removing the live roots and replacing the soil inside a metal 13 cm core
with three large root windows. Two ingrowth cores were recovered from an area within 1m of each minirhizotron (i.e. within the same microhabitat) at each date using a soil auger (total n = 72). Soil cores were 4.5 cm wide and 30 cm deep in December 2016 (n = 36, one per minirhizotron) but reduced to 20 cm depth (n = 108, 3 per minirhizotron) on subsequent samplings. In this manuscript we only consider the top 13 cm, to pair with root ingrowth cores. The December 2016 set of ingrowth cores was amended with a separate amount of replacement litter designed to equalize the total root litter to a previously observed
seasonal site mean. This was part of a separate isotope labelling experiment not detailed in this paper. Root decomposition rates in dehesas are typically very high (Casals et al., 2010) and when combined with a relatively coarse threshold for sieving we judged that these data were acceptable to use along with unamended cores as most added root litter would be fragmented by the time of sampling. In all cases, roots were extracted from soil samples by passing through a 2 mm mesh, and picking through the remaining material for intact roots. The extracted root material was cleaned in distilled water and dried at 40°C
until weight loss ceased. The final weight was recorded as root biomass.

In addition we made measurements of extractable N and P content in surface soil during the period of the experiment. These were not from the same cores as root measurements but closely paired (< 1m) at the same sampling location. Two thirds of the minirhizotron and root sampling locations were used with an even distribution of habitats and treatments. Soil from 3, 0-5

cm cores was bulked and sieved (2 mm) and stored overnight at 4 °C. Sub-samples of 20 g were extracted by shaking for 1 hour in 100 mL of 2M KCl (for inorganic N), or 0.5 M NaHCO$_3$ for (phosphate-P), then filtering the supernatant through Whatman no. 1 (N) and no. 42 (P) filter papers that were preleached with 30 mL distilled water. Extracts were analyzed for ammonium, nitrate, and phosphate using standard colorimetric methods on a Lachat QuickChem 8500 (Lachat Instruments,

Hach Company, Loveland CO, USA). Additional 7 g sub-samples were oven dried for 48 hours (until mass loss ceased) at 45°C to determine gravimetric water content.

## 2.6   Comparisons with site-level and above-ground measurements

We also compared root profiles both a) over time and b) with depth to site-level instrumentation. These were i) harvests of above-ground understory biomass made in each treatment area, dried to determine dry weight on 23 March 2017 and 25 May

2017, used to calculate root:shoot ratios by comparison with the direct biomass measurements made at comparable times and ii) phenocam derived green chromatic coordinate (GCC) (Luo et al., 2018) and normalized difference vegetation index (NDVI) of the herbaceous layer measured with an infrared enabled digital camera (StarDot technologies, USA). These two properties are common methods of assessing above-ground phenology from proximal remote sensing. In both cases these above-ground measurements were not directly paired with root measurements, so root:shoot ratio and phenological synchronicity is examined

on the treatment-level.

## 2.7   Statistical Analyses

All statistical analyses were conducted in R (R Core Team, 2018), version 3.50.

As previously mentioned, in this study, we assumed that all measurements at individual locations containing paired minirhizotron, root cores, and soil measurements were independent, given the large distance between locations and pretreatment

similarity. We modelled the distribution within treatments as a nested structure, so that the sets of four 'microhabitats' (individually either located in pasture or under oak canopy) was associated within a particular central tree within a particular nutrient (control, +N or +NP) treatment. Due to the lower number of replicates in soil measurements, we removed this nested term to avoid overfitting the statistical models used for these data. As individual minirhizotrons were always at least 5 m apart and arranged linearly, we expected this spatial co-variation associated around any particular tree to be minimal. This was especially

true between the pasture microhabitats, on opposite sides of the tree, where the biggest treatment-level differences tended to be found.

We used the R package lme4 (Bates et al., 2015) to fit mixed effect models and performed a series of linear and generalized-linear regressions, aiming to explain i) relationship between image RCI with image RLD, volume per area or root number in order to validate this fast markup and ii) use observed differences between treatments (control, +N, +NP) and microhabitats

(tree canopy and open pasture) to explain root dynamics (both from minirhizotrons and direct soil cores) in this system. For the validation of the mark-up methodology we tested different empirical models (linear to polynomial) and we chose the best model for further analyses using BIC model selection, bearing in mind that the granular RCI measurement would always fall within a range of 0-100 so polynomial models with inflection points outside this range could be valid for conversion between

RCI and RLD. Data was transformed where appropriate to meet the assumptions of models used, primarily by transformations using Tukey's ladder of powers using the rcompanion package (Mangiafico, 2018).

We used mixed effects models to understand the effect of nutrient and habitat on both minirhizotron observations (predicted RLD) and soil measurements (root biomass, soil extractable nutrients). In these, we assumed that individual sampling loca-
tions and sampling dates were crossed random effects but did not implement a time series correlation structure due to the large interval between observations and the rapid turnover of species (Fernández-Moya et al., 2011) throughout the year. We performed simple linear model comparisons within individual sampling dates, reducing the models to their most parsimonious form via Akaike Information Criterion (AIC). We tested for appropriateness of including interaction effects between treatment and location by comparing models with and with-out interaction terms, checked residuals for normality, and report P values
for these models using Satterthwaite approximation of degrees of freedom (Satterthwaite, 1946).

## 3   RESULTS

### 3.1   Treatment Effects on Soil N:P stoichiometry

During the period of this study there was a strong effect on the ratios of extractable N and P measured in surface soil. Extractable inorganic N content (Figure S1) was significantly higher in tree canopy (($2.37 \pm 3.8$ (sd.) mg g$^{-1}$) than pasture ($1.79 \pm 3.1$ mg
g$^{-1}$) microhabitats ($P < 0.05$) but only had a marginally significant effect from the treatment +N ($P = 0.07$, driven by very high N contents in some samples in March 2017). Olsen-extractable P (Figure S2) was very different between both treatments and habitat type, with more P available in the +NP treatment ($P < 0.001$) and in pasture microhabitats ($P < 0.001$). In the control and +N treatment, mean extractable P over the period of the experiment was $3.00 \pm 2.78$ $\mu$g g$^{-1}$ in tree covered microhabitats and $1.44 \pm 0.9$ $\mu$g $^{-1}$ in pastures. When P was added in +NP, these phosphate-P concentrations were $7.03 \pm 5.6$ $\mu$g g$^{-1}$ in tree
covered microhabitats and $3.5 \pm 1.54$ $\mu$g g$^{-1}$ in pastures. Overall these differences led to a strong difference in the ratio of bioavailable N and P for the treatment types ($P < 0.001$, Figure **??**), with a higher ($P = 0.06$) ratio in +N and a lower ($P < 0.001$) ratio in +NP than the control. This difference was bigger than the habitat effect ($P = 0.08$) on available N:P.

### 3.2   Validation of Minirhizotron Markup

We found a good correlation of our markup method at our site compared to all Rootfly-derived indices. RLD, volume per area,
and root number were well predicted using the fast cover markup with the best predictive models (3rd order polynomials (RLD, vol area), 2nd order polynomials (root number) using BIC model selection) having $R^2$ of 0.77 (Figure 3) and 0.78, and 0.67 respectively. Particularly high root density caused saturation in RCI but not in these measurements, but as this affected only a small number of images in the validation dataset and residuals were otherwise normally distributed, we converted treatment mean RCI to RLD using the observed relationship in all further analyses.

### 3.3 Minirhizotron-Derived Root Length Density

In general, the minirhizotron images contained less roots in tree covered than pasture microhabitats (across the whole dataset, $P < 0.001$, Figure 4). RLD decreased with depth at all periods of the year (Figure 5), with the deepest soil (40 cm) having a mean RLD of $0.07 \pm 0.01$ (S.E.) mm mm$^{-2}$ at its highest and $0.02 \pm 0.01$ in the least abundant period (December). This peak in deep soil was in May while the overall seasonal cycle (including above-ground biomass, shown later) peaked in March. The seasonal cycle was most evident in the surface soil, where maximum RLD was $0.50 \pm 0.03$ mm mm$^{-2}$ in pastures, and $0.40 \pm 0.03$ mm mm$^{-2}$ in tree covered microhabitats. Following March, root biomass declined through May to September (mean $0.05 \pm 0.01$ mm mm$^{-2}$ in pastures, $0.04 \pm 0.01$ mm mm$^{-2}$ tree covered), and stayed relatively constant until December (pastures: $0.07 \pm 0.01$ mm mm$^{-2}$, tree covered: $0.06 \pm 0.01$ mm mm$^{-2}$).

RLD was higher in all measured depths in the +NP plot during the growing season, peaking in the March 2017 sample (Figure 5). Taking the cumulative RLD in the top 13 cm of soil (corresponding to the depth of our ingrowth cores and containing the majority of roots, we compared the treatment x microhabitat effect on RLD. Both microhabitat ($P < 0.005$) and +NP ($P < 0.05$) had significant effects on the RLD calculated from the minirhizotrons over the experiment, but the +N treatment did not differ from control ($P = 0.33$). Differences between the nutrient treatments were smaller than variation between microhabitats or within time but evident in some periods of the growing season. This difference tended to be larger during the spring as low average RLDs outside the main growing season meant absolute differences between nutrient treatments, if they existed at this time, were impossible to detect using our methodology.

### 3.4 Root Biomass and Root Ingrowth Measurements

The two methods of direct soil measurement (soil cores and ingrowth cores) produced similar results, indicating a seasonal cycle of root biomass similar to that measured by the minirhizotrons (Figure 6). The top 13 cm root biomass in December 2016 (tree covered median 2020 kg ha$^{-1}$, pasture median 1140 kg ha$^{-1}$) had no significant treatment effects but a difference between microhabitats ($P < 0.05$). As the ecosystem developed into the spring growing period there was a difference in these absolute stocks in March 2017 (tree covered median 6390 kg ha$^{-1}$, pasture median 5670 kg ha$^{-1}$). Here +N had significantly more ($P < 0.05$) roots than +NP and control. This treatment difference was strong enough that the most parsimonious (AIC) model at this date did not include a microhabitat effect unlike all other comparisons

From the ingrowth cores, we also found significant effects. The most parsimonious model for recovery in May 2017 found an effect of both nutrient treatments, where both nutrient amended treatments increased over control ( $P < 0.05$) and also significantly less production in pasture areas compared to those under tree canopies ( $P < 0.001$) but no interaction. For the year-round ingrowth cores the treatment effect was lost in December 2017 but the highly significant ($P < 0.001$) microhabitat effect remained. Likewise, the most parsimonious models showed that in March 2018 (where an interaction term remained in the model), +N had significantly more root biomass in the cores ( $P < 0.001$) and differed between microhabitats (+N-microhabitat interaction, $P < 0.01$). In May 2018, both microhabitat ($P < 0.01$) and both nutrient treatments (+N, $P < 0.05$; +NP, $P < 0.05$) had significant effects. Post-hoc Tukey HSD groupings for linear models for all individual dates are shown in

Figure 6.

## 3.5  Pairing of Biomass Dynamics Above- and Below-Ground

The short-term minirhizotron measurements in May 2017, separated by 10 days around a rain pulse (1 day pre-pulse, 6 days post-pulse) showed a clear proliferation of roots following the pulse (Figure 7). This increase was significant ($P < 0.001$) in both pasture and tree-covered microhabitats and evident in all soil depths measured with the minirhizotron. Similar short-term responses were evident in NDVI and GCC during this period (Figure 8). The relatively sparse distribution of minirhizotron campaigns means we were unable to diagnose similar responses to other rain events although from these site-level above-ground indices the May event was the largest shift against the overall trend for the year. From comparison with site indices it is also notable that while the minirhizotron root-cover time series correlates well with both NDVI and GCC in respect to the March peak and decline into the summer dry period, root cover was not in sync with either of these indices in autumn. In both 2016 and 2017 winters, RLD was low but had recovered by the March of the following year (Figure 8). This suggests that the majority of root growth was in the period of December to March unsampled in either year by the minirhizotron campaigns and after the apparent 'green up' of the ecosystem from near-surface remote sensing.

## 3.6  Root:Shoot Ratios

For the two campaigns where above-ground biomass data was available for the herbaceous layer, even only using the top 13 cm of soil indicated that root:shoot ratio was very large (in control treatments in March 2017, these were 20:1 (OP) and 15:1 (UC), while in May 2017 these were 21:1 and 22:1). Nutrient treatments showed a typically higher ratio in the +N compared to +NP treatment in March although by May this difference had been lost from pastures. Generally, root:shoot ratio was higher in +N than control but equal or lower in +NP than control (Table 2). Regardless of this potential change in root:shoot ratios between treatments, the magnitude of the difference between amount of roots and shoots across all treatments was substantially larger than any changes in root biomass induced by our nutrient treatments.

## 4  DISCUSSION

Mediterranean systems are heavily influenced by seasonal climate and as expected we found strong seasonal effects on root dynamics. Likewise, differences were evident between tree and pasture microhabitats but interestingly, the direction of the effect reversed between RLD (measured from minirhizotrons) and root biomass (measured directly) (Figure 4, 6). The nutrient treatments in the experiment also had effects on the amounts of roots measured which differed betweeen RLD and absolute biomass suggesting that effects were not just higher productivity when more nutrients (+N,+NP) were available. Measurements of roots must always be carefully interpreted (Mancuso, 2012) as all procedures are both affected by methodological biases and subject to logistical constraints. Hence before discussing nutrient treatments in depth, we will briefly address the interpretative trade-offs between methods.

## 4.1 Methodological Considerations

Minirhizotrons are non-destructive measurements but require the presence of an observatory. Artefacts due to observatory presence are particularly acute close to the time of installation (Joslin et al., 2001) and there is little consensus towards an appropriate time to leave observatories before good data is collected (see Johnson et al., 2001; Mueller et al., 2018; Strand et al., 2018). Slow-growing species may also take considerably longer than this time to equilibrate (Strand et al., 2018) but most (perennial) *Q. ilex* roots probably reached deeper soil layers than our observations (Moreno et al., 2005), causing both of our methods to mostly sample herbaceous layer roots. The eight months before first measurements included summer drought and almost total annual senescence of this herbaceous layer, followed by autumn rewetting (as shown in Figure 4). We expected this to have stronger effects than observatory presence on root growth around minirhizotrons and so time since installation was unlikely to have impacted the observed trends.

Rapid processing of roots from soil cores does not allow architectural properties to be easily examined in dry systems, as roots are often fragmented during sieving and breaking up of soil clusters. On the other hand, minirhizotron measurements do not alter the position or distribution of roots once they have colonized the area around the tube. Our processing method for minirhizotrons (which was calibrated for our site only) validated well against most of the range of data (Figure 3, $R^2 = 0.77$), with a polynomial fit due to the highest root density being smaller than the resolution of our markup. Using our direct methods, we treated all root biomass remaining within the sieve as roots, while the visual minirhizotron method allowed roots to be ignored if broken or clearly dead. Excluding this possibility of misdiagnosed root biomass in soil cores at different periods of the year (i.e. more root litter later in the season), we found that RLD decreased from March 2017 to May 2017 and root biomass increased slightly. A similar, but smaller difference was observed in 2018. We assume that this difference was due to difference in responses of root traits such as RLD when compared to trends in traits such as root biomass / diameter between treatments and habitats through seasonal changes in weather and water availability.

Similarly, another potential source of errors in our study is the effect of low replication, particularly acute in systems such as our study site, which are highly diverse and heterogeneous (Moreno, 2008). Below-ground systems additionally cannot be seen before sampling and representative locations are often assigned based on above-ground properties. We used 6 replicates per nutrient-vegetation combination for minirhizotrons (36 in total). This level of replication was similar to other multifactorial field experiments using minirhizotrons (Ziter and Macdougall, 2013; Arndal et al., 2017; Mueller et al., 2018) and, we accounted for consistency in resampling microsites in the statistical models for root data. We hence treat the differences shown by different methods as being both real and ecologically relevant for the rest of this discussion.

## 4.2 Effectiveness of Nutrient Treatments

The nutrient treatments used at each site were designed to induce a N:P stoichiometric imbalance with N addition and reduce it by restoring N:P ratios when adding +NP (El-Madany et al., 2018). The soil sampling during this experiment (Figure **??**, S2,S3) indicated that the intended stoichiometric differences between treatments were maintained, particularly in the case of +NP, where the increased bioavailable P from the fertilizer (Weiner et al., 2018) was still evident across both treatments and

habitat types during the period of this study. Therefore differences in +N and +NP treatments together can be interpreted as N effects and differences in +N alone can be interpreted as P deficiency, while +NP effects above control without corresponding N effects indicate co-limitation of the nutrients together. As this study was conducted several years after fertilization, and N is generally more prone to ecosystem losses than P, the overall trend in decreasing ratios in +NP, and less strong increases in +N agreed with our expectations.

## 4.3  Treatment and Microhabitat Effects

We expected root production (which we measured in terms of RLD and biomass) to differ between both habitat types and nutrient treatments at our site. Tree-grass systems, combining short-term and long-term optimality in vegetation habit (Eagleson and Segarra, 1985), usually occur in areas with major seasonal variation in water availability, leading to major variation in soil and understory properties between microhabitats (Moreno et al., 2013). These typically include altered soil water storage (Joffre and Rambal, 1993), and water stress (Joffre et al., 1987). Trees increase shading, allowing reduced transpiration beneath the tree, while extending their root systems to obtain water even from areas outside the direct influence of their canopies (Cubera and Moreno, 2007). Additionally, litter input and waste from animals congregating beneath trees result in higher soil organic carbon (Howlett et al., 2011), N (Gallardo, 2003) and sometimes P (Gallardo, 2003; Rolo et al., 2013) beneath trees and in many cases lead to higher herbaceous layer above-ground biomass (Moreno, 2008; Rivest et al., 2011). Our site had higher C, N and P contents between tree- and grass- areas (Table 1) which corresponded with higher total above-ground biomass in March 2017 (Table 8) and root biomass (Figure 6) throughout the experiment beneath canopies than in pastureland. A positive effect of tree on above-ground yield has been reported in many tree grass systems (e.g. Puerto, 1992; Frost and McDougald, 1989). However in May, we found greater above-ground biomass in pasture microhabitats (Table 2) compared to those covered by tree canopies. This inversion of microhabitat effect in late Spring above-ground but not belowground in pasture microhabitats was possibly due to differing grazing pressure or plant responses to water availability; the tree-covered microhabitats probably depleted the water in the summer drying faster than in open areas (Moreno and Cubera, 2008; Moreno, 2008), which resulted in more root production for water uptake compared to above-ground growth (and increasing root:shoot ratios as observed in Table 2). This seasonal shift in observed effect highlights the importance of seasonal observations to understand biomass dynamics in such systems.

The same difference in roots between habitat type was not found from minirhizotrons, where we consistently observed greater RLD in open pasture throughout the year. Between tree-covered and pasture areas of dehesas there are major differences in herbaceous layer diversity (López-Carrasco et al., 2015) and community vegetation composition (Marañon, 1986), which affects plant trait distributions. RLD and specific root length are important belowground plant traits linked to plants ability to explore soil and acquire resources (Fort et al., 2014) thus potentially important to competitive success in herbaceous layer communities. While our pre-treatment data available for this study indicated little differences in basic soil properties between habitats (slight differences in texture), surface soil measurements (Table 1, Figures **??**, S1, S2) suggest that soil conditions were different in these two habitat types, potentially promoting more RLD-producing species in the open pasture.

Alongside these induced changes, and primarily in the most productive parts of the year, we observed increased RLD in +NP ($P < 0.05$) compared to the N and control treatments (Figure 4), particularly in open pasture microhabitats. The direct biomass (pooled from direct measurements and ingrowth cores, Figure 6) showed a stronger effect of +N ($P < 0.05$ over the whole experiment) than +NP on total root mass recovered, particularly under canopies. Biomass and other root traits and properties

such as root length may be affected in dissimilar ways by the nutrient and habitat treatments. This is especially important for the uppermost soil layer, where fine roots appeared to disappear progressively with soil dryness leaving remaining roots to grow in diameter to support developing water-harvesting architecture in deeper layers (and hence, a decrease in RLD in May but an increased or similar biomass, Figure 4,6). Stoichiometric variation provoke modulation of these responses in terms of root system architecture and traits (Drew, 1975) , potentially due to either a change in species community or in traits themselves

(e.g. root diameter). Unlike N ions, P is relatively immobile in soil, so in general, P availability promotes primary root growth at the expense of lateral development (Williamson, 2001). Less surface soil exploration is necessary under high availability of immobile P, reducing the need to access P-rich plant residues (Lynch, 2011). P addition hence promotes development to reach deeper soil layers and water-soluble nutrients (e.g. nitrate). Thus, a proliferation of roots in the topsoil in the P-limited +N treatment but more primary roots (and hence, root length from the minirhizotrons) in +NP grasslands alongside a general

increased production as a result of the nutrient additions can explain the difference between treatments. The observed difference between +NP and control in RLD, despite the intended difference in stoichiometric conditions may be due to the greater loss of added N than P since the start of the experiment. The available N and P (Figure **??**) suggests that in +NP, ratios were actually lower than the control (i.e. relatively more P available than N) which may have prompted this increased production of RLD in search of additional N from deeper in the soil.

An increase in primary roots and increased RLD from grasses may be easily detected by the minirhizotrons, as primary roots follow areas of disturbance, previous root channels or soil objects (Rasse and Smucker, 1998) to allow penetration of deeper soil. Proliferation in surface soil layers already heavily colonized by roots is more difficult to measure using our root presence method. Likewise, our method did not allow an assessment of thickness of individual roots, which may indicate adaption for water uptake. This explanation is supported by the observation from minirhizotrons that roots shifted towards deeper locations

late in the growing season (Mar-May 2017, Figure 5). Root biomass declined in drying surface soils at the onset of summer but production continued in deeper areas, presumably where water was still accessible. Both changes in root diameters and dense proliferation would potentially increase root biomass, as observed in the ingrowth cores, without necessarily changing the RLD observed from the minirhizotrons. This seasonal shift could be due to individuals or shifts in species composition; little information on specific species root phenology in dehesas is available, but it is clear that traits relating to root growth seasonal

timings exploratory and water properties can differ substantially both between species (e.g. Luke McCormack et al., 2014; Fitter, 1986) and in time. A better pairing between difficult to measure root traits and relatively accessible above-ground plant traits may allow a more diagnostic understanding of root behavior under a highly diverse (Moreno et al., 2016) above-ground system and further differentiation between these treatments.

The trait effect observed from nutrient treatments would also explain the reversal of directional effects between habitats when measured with minirhizotrons and cores. Similar experiments at our site have previously reported changes in canopy functional traits in response to nutrient addition (Migliavacca et al., 2017) but cannot diagnose changes belowground. A greater abundance in 'high RLD producing' species in pastures, where RLD and soil exploration is a critical competitive trait may have led to a greater response for the minirhizotron observatories. High RLD is potentially a strong nutrient uptake strategy if roots are cheap (i.e. high specific root length (Hodge, 2004)) and capable of high nutrient uptake. Interestingly, tree cover tends to promote more (relatively high RLD) graminoids over legumes and forbs in dehesas (López-Carrasco et al., 2015), and N abundance (comparatively higher under canopies in dehesa systems (Gallardo, 2003)) tends to shift community composition towards grasses (Bobbink et al., 2010). However, in heterogeneous environments, high RLD may provide competitive advantages, even if high RLD individuals are less focused on particular nutrient hotspots (Mommer et al., 2011). Hence the relatively more homogeneous environments under canopies with thicker organic layers and more abundant background levels of nutrients may mean that nutrient-searching ('high RLD') strategies are less important compared to grassland areas even if nutrient re-cycling through recalcitrant litter means root biomass is higher, as observed.

## 4.4 Changes in Root:Shoot Ratio

Conclusions about a +NP effect on root architecture rather than an absolute increase in biomass compared to +N are supported by general increases in (biomass-based) herbaceous layer root:shoot ratio (Table 2) in +N, but not +NP. Compared to the other treatments, this ratio of pasture plant biomass decreased in +NP, as expected with a decreasing plant biomass investment belowground with greater supplies of both nutrients, despite the increasing root biomass. With increased biomass, but 'ambient' stoichiometric ratio of these elements, the plant community appeared to invest less biomass belowground. The only large increases in root:shoot ratio were in +N, potentially indicating that herbaceous plants were producing relatively more roots to alleviate the induced P limitation. Nutrient stress may increase root:shoot ratio if these deficits are the major growth constraint (Erikson, 1993; Ågren and Franklin, 2003) and can occur on the ecosystem and individual plant level (Fichtner and Schulze, 1992). This suggested that while the overall system may have shifted towards higher RLD species in the +NP treatment, the system was more 'nutrient stressed' under +N, as soil N:P diverged from ambient conditions (Figure **??**). Notably, from the two dates where we could calculate root: shoot ratio, the difference induced by +N was not found in May in open pasture microhabitats. As with the overall biomass responses, this was potentially due to the effect of the dry down in more exposed areas away from tree canopies, where the herbaceous layer was already in decline.

Generally, the root:shoot ratios (15-30:1) in this experiment were very large. This was driven by a high root biomass, peaking at around 8000 kg ha$^{-1}$ despite being calculated based on only the top 13 cm. Other studies in similar landscapes in southwestern Europe found maximum root masses of around 2000 kg ha$^{-1}$ (dependent on cover type; Rolo and Moreno, 2012) , 2500 kg ha$^{-1}$ (Jongen et al., 2013) or under in nearby walnut forestry, 300-400 kg ha$^{-1}$ (López-Díaz et al., 2017), where pasture production was around 2 x higher than our site (G. Moreno, pers. comm). Despite sampling soil profiles to deeper depths, root:shoot ratios at these sites were considerably lower than ours (e.g. 4:1 Jongen et al., 2013) and more consistent with global means ('temperate grassland' mean 4.2:1, and 'savanna' mean 0.6:1 (Mokany et al., 2006)). We are confident in the magnitude

of root masses reported in this study due to their consistency both over time and between treatments (Figure 6). In seasonally dry systems, biomass allocation may shift towards roots due to multiple factors (Chapin et al., 1993). Root distribution in these systems tend to be shallow and wide (Schenk and Jackson, 2002), and grazing of above-ground vegetation (McNaughton et al., 1998) both reduces transpiration water losses (through reduced leaf area) and requires root foraging for nutrients to replacing this removed biomass. As the year progressed towards the summer drought and grazing pressures increased, root:shoot ratio shifted towards roots, and root profiles became more evenly vertically distributed (Figure 5), agreeing with this explanation of the ratios at the site. Ecological effects of seasonal and fertility-related changes are also likely heavily modified by annual variation (Vaughn and Young, 2010) in weather, and the particular conditions of the 2017 growing season (a dry year following a particularly productive previous year at the site (Luo et al., 2018)) may have contributed to these high ratios. However, similar root: shoot ratios to ours have been reported earlier in the season (Puerto, 1992) at dehesa sites. More frequent sampling is necessary over multiple years to disentangle whether these ratios are representative of the rest of the growing season.

## 4.5    Seasonality of Root Biomass and Linkage to Above-Ground Phenology

Our site has a highly seasonal climate with severe deficits of water in summer and an excess in winter (Perez-Priego et al., 2017) and hence short-term root dynamics are particularly interesting. In the control treatment we observed that root growth responded quickly to a short-term rain event in the late growing season (Figure 7) as RLD increased in all soil depths following a rain pulse in May. This event was paired by a clear response in both NDVI and GCC interrupting the general decline in the late growing season dry down as both shoots and roots responded near-simultaneously. However, measurements in autumn implied a desynchronization between above- and below- ground during the early growing season in both years studied. Most of the root production appeared to occur after measurement in December and before measurement in March, indicating that the key periods of root production were overwinter rather than early or late in the growing season. The initiation of major periods of root growth had not begun by December in either year, while both GCC and NDVI of grassland areas had reached growing season levels by this point in all treatments (Figure 8). This difference was presumably due to high water availability but decreasing light availability in autumn leading to prioritizing of above-ground LAI development, compared to decreasing water availability but abundant light in spring. As both GCC and NDVI are commonly used to track plant phenology (e.g. in a semi-arid grassland, Browning et al., 2017), this was particularly interesting as grassland systems are expected to be highly synchronized above- and below- ground (Steinaker and Wilson, 2008). This difference was substantially larger than the 2-4 weeks observed in other grasslands (Steinaker et al., 2010), and, coupled with the high root:shoot ratio we observed during the spring, indicates that overall productivity and coupling between ecosystem-level productivity and respiration depends largely on the below-ground system at our site. Additionally, unlike the four Mediterranean studies in Abramoff and Finzi (2015)'s meta-analysis, this desynchronisation also indicated that leaf growth was before root growth in our system. This may be linked to the very severe summer drought with extensive root systems being more important for water uptake in the late growing season than nutrient uptake for biomass production in the early growing season. On the other hand, the tight coupling of above- and below- ground vegetation in May (during the dry down late in the growing season) indicated that at some points in the

year these dynamics are highly coupled in the short term. This is likely when water stress is high; while our minirhizotron data was not as high resolution at other times, seasonal effects did not appear to be so closely tied to rain pulses above-ground earlier in the year (Figure 8). At our site, the expectation of root production in direct synchronicity with shoot production is clearly dependent on seasonal and /or climatological conditions. High resolution measurements are necessary to capture such events and advancements in minirhizotron technology including autonomously operational, frequent image capture (matching above-ground proximal remote sensing) and methods to analyze these data, will allow much greater understanding of seasonal cycles in root production and their link to above-ground productivity.

## 5 Conclusions

Much of our ecosystem-level understanding of plant seasonality and responses to global change is drawn from above-ground measurements but it is not clear how well this understanding holds belowground. In the highly seasonal system of this study, we found the cycle of root dynamics broadly matching both the above ground 'growing' season and seasonal patterns of above-ground biomass as inferred from near-surface remote sensed measurements. However there was a notable delay in root production in the early growing season with most root growth not commencing until well after the regreening of the system. N addition increased root biomass and root shoot ratios in productive periods for herbaceous layer vegetation. +NP (with similar N:P availability to control) showed increases in RLD but not increases in root biomass. This indicates that nutrient availability-driven changes and stoichiometry-driven changes are not necessarily the same in our system. The expected gradual shift towards P limitation with N deposition could therefore drive increased plant allocation belowground on the community level but also alter traits and function in ways dependent on other environmental conditions, including N:P stoichiometry.

Further work should focus on i) understanding community and plant trait responses to ecosystem stoichiometry, especially in highly diverse and seasonal communities with a large pool of species able to exploit changes ii) understanding the consequences of these responses to ecosystem function and iii) increasing temporal density of belowground observations. Phenology responses to global change factors may be on the scale of days and hence high frequency sampling is essential to understand the belowground response to such forcings and the mechanistic effects of global change on such communities.

*Data availability.*

The authors are eager to share the minirhizotron imagery used in this study with groups working on the critical problem of interpreting minirhizotron imagery. However the large size of the datasets makes permanent hosting of these images difficult. Please contact the authors for distribution of these data.

*Author contributions.* RN designed the experiment, performed the root-associated field, lab and data analysis work, and wrote the manuscript. KM performed all soil nutrient extractions and respective sample collection. MH designed the minirhizotron system. YL provided the pro-

cessed proximal remote sensed data and assisted with their interpretation. GM provided both above-ground data, and fieldwork assistance. MS and MM helped design the experiment, all authors helped in manuscript preparation.

*Competing interests.*   No competing interests are present.

*Acknowledgements.*   This work was funded by a Marie Skłodowska-Curie Individual Fellowship to RN and the via the Alexander von Humboldt Foundation Max Planck Research Prize to MR). We acknowledge the invaluable field support of Enrique Juarez, Andrew Durso, and Jinhong Guan as well as Mónica Lorente, Fatima Khalid, and Rolf Rödiger who assisted in laboratory processing of root samples. We are also especially grateful to Gianluca Fillipa for sharing his expertise with interactive GUIs for graphics processing in R.

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

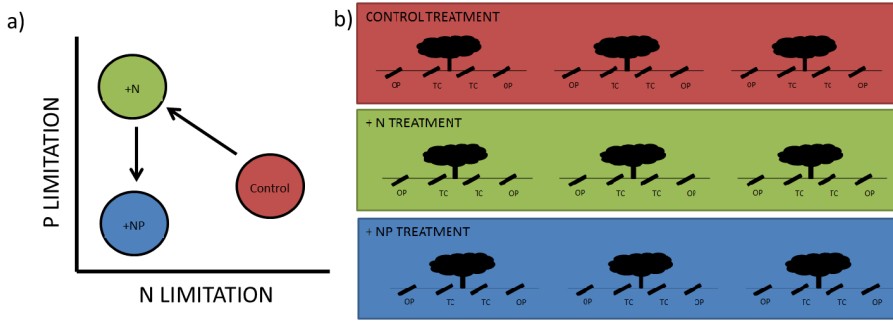

**Figure 1.** Schematic diagram of experiment. In panel a), we show conceptual release from N limitation in our +N treatment but promotion of P limitation which is alleviated by NP additions. In panel a) we show 12 minirhizotrons per treatment, split between three trees into open pasture ('OP) and tree covered ('TC') areas. This is a small subset of the total trees in each footprint. Median distance between trees within treatment is 116 metres and minimum distance between individual minirhizotrons is 5 metres.

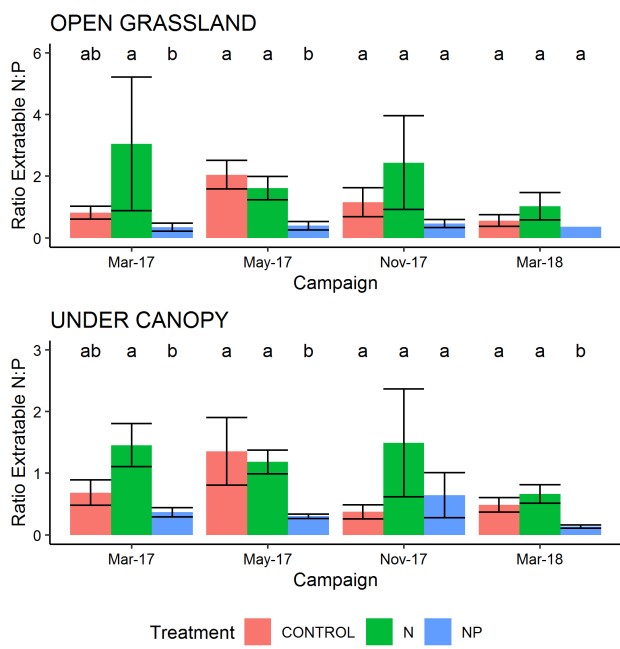

**Figure 2.** Ratio of extractable inorganic N (2M KCL) to extractable P (Olsen method) in 0-5 cm soil during the experiment. Over all measurements, the +N treatment has a marginally significant (P = 0.06) greater N:P ratio and the +NP treatment a significantly lower (P < 0.001) ratio than control. Letters show Tukey HSD groupings within campaign-habitat combinations and errorbars show standard error of the mean.

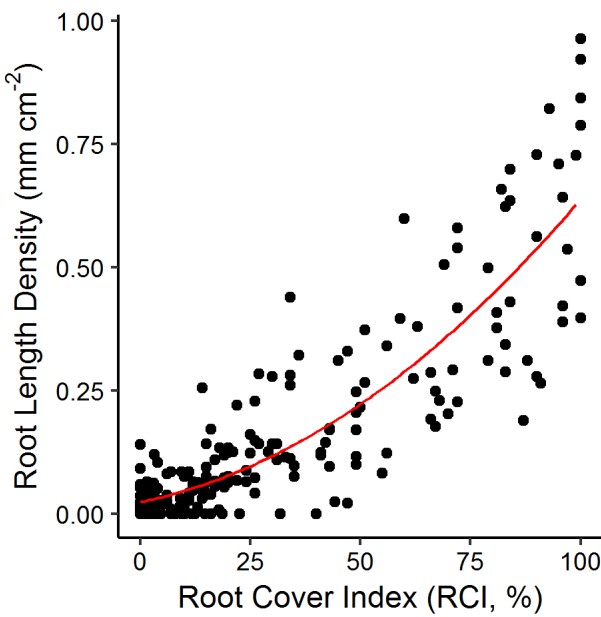

**Figure 3.** Root Cover Index (RCI) against root length density (RLD) for a random sample of images (n = 300) from three imaging campaigns at our site. A 3rd order polynomial (indicated) fit to these data with an $R^2$ of 0.77 was used to predict RLD for all other figures.

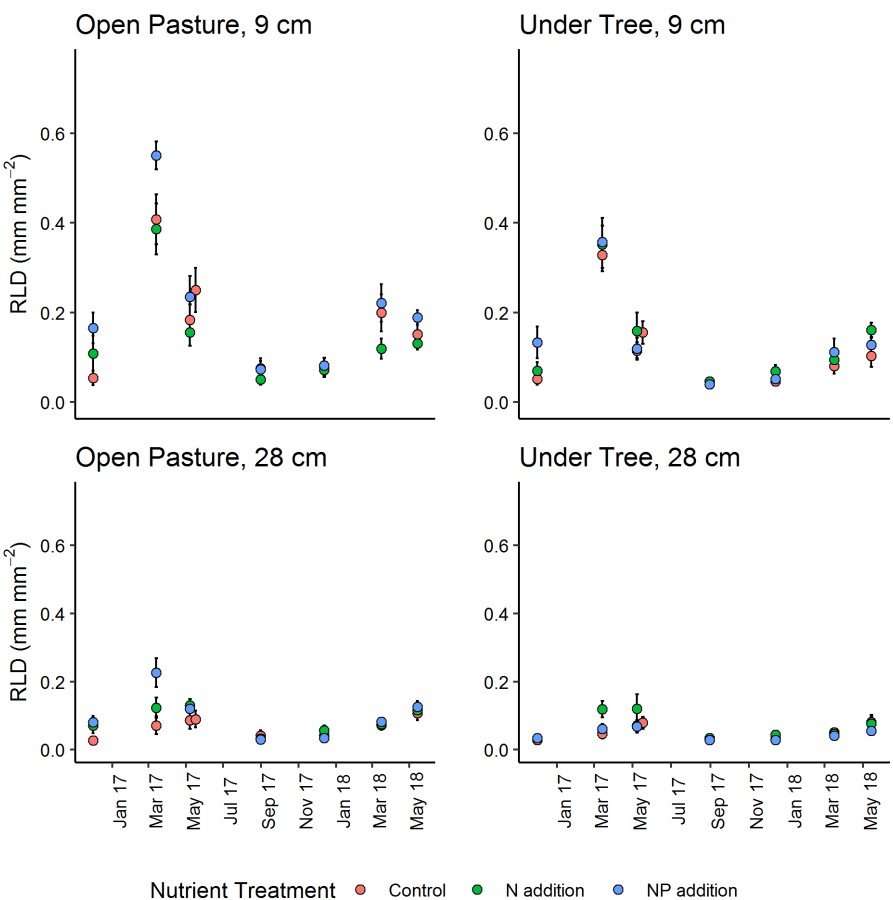

**Figure 4.** Seasonal cycle of minirhizotron-calculated root length density (RLD) between treatment at two selected depths (vertical position of camera). +NP treatments tended to diverge, especially in spring 2017. Error bars show value ± SE. A mixed effect model revealed a significant effect of the +NP treatment (P < 0.005) but N did not differ from control.

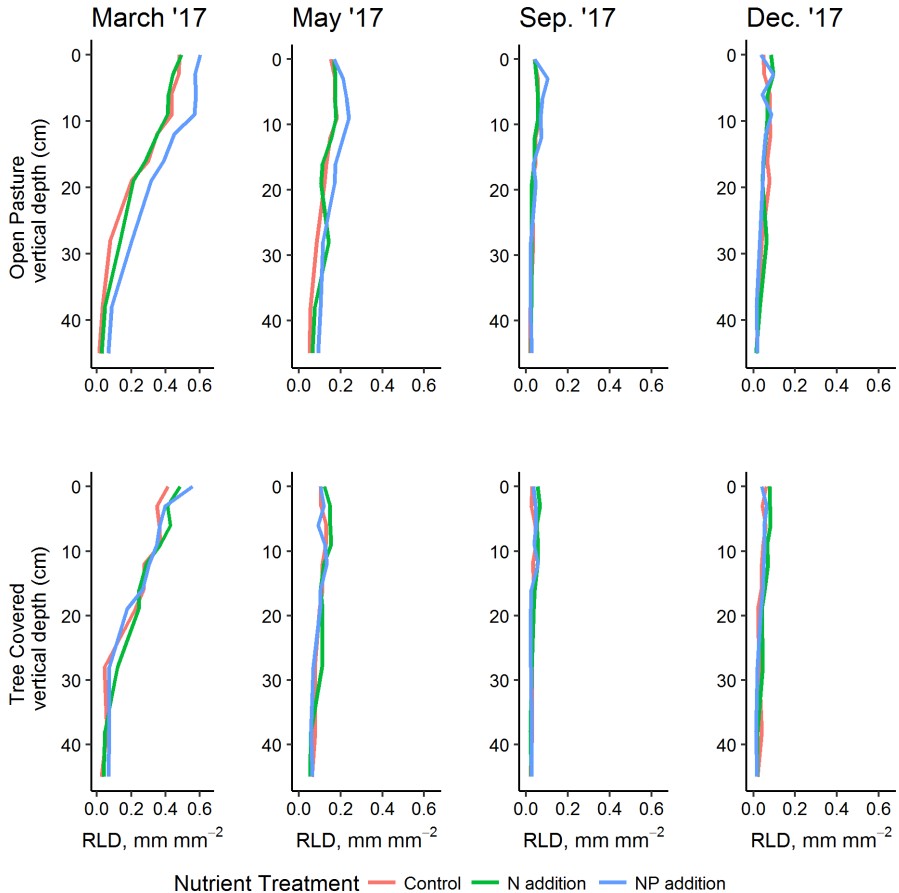

**Figure 5.** Vertical root profiles (mean of minirhizotron measurements) for the four sampling dates in 2017, representing one complete annual cycle (peak RLD in March of this year was also higher than 2018). +NP treatments tended to diverge in the grassland, especially in surface soils where RLD was highest. Treatment effects were most evident in the shallow soil depths where root biomass was highest.

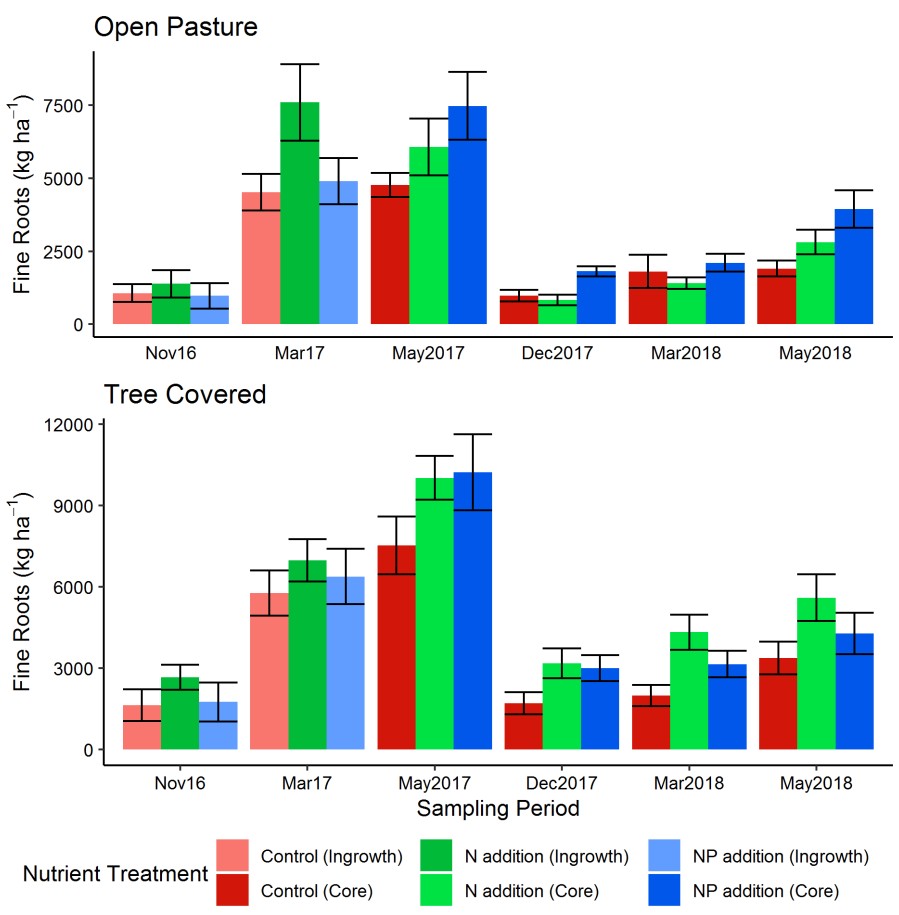

**Figure 6.** Fine root biomass in top 13 cm from direct soil cores (Nov. 16, Nov. 17) and ingrowth cores (installed Dec.2016, removed in Mar.17, Nov.17, then installed Nov.17, removed Mar.18, May18). Root biomass showed a seasonal cycle and also differences between treatments, with more roots in general under canopies and in fertilized plots. Error bars show value ± SE and letters indicate Tukey-HSD groupings for most parsimonious models within treatment for both open pasture and tree covered microhabitats combined in individual sessions. Both +N and +NP treatments tended to have more root biomass than control treatments. Across the whole dataset, +N had significantly (P < 0.05) more fine roots than the other two treatments.

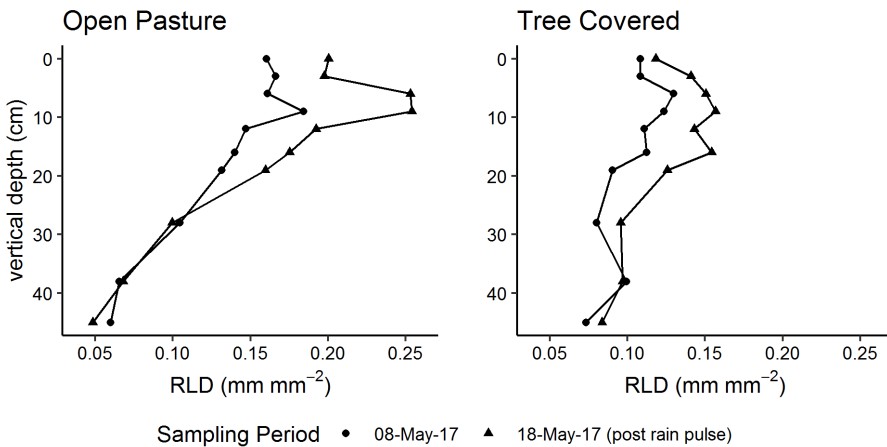

**Figure 7.** Control Treatment root length density (RLD) response to rain pulse in May 2017. RLD throughout the soil profile (P < 0.001 for both microhabitats) indicating short-term proliferation of root growth.

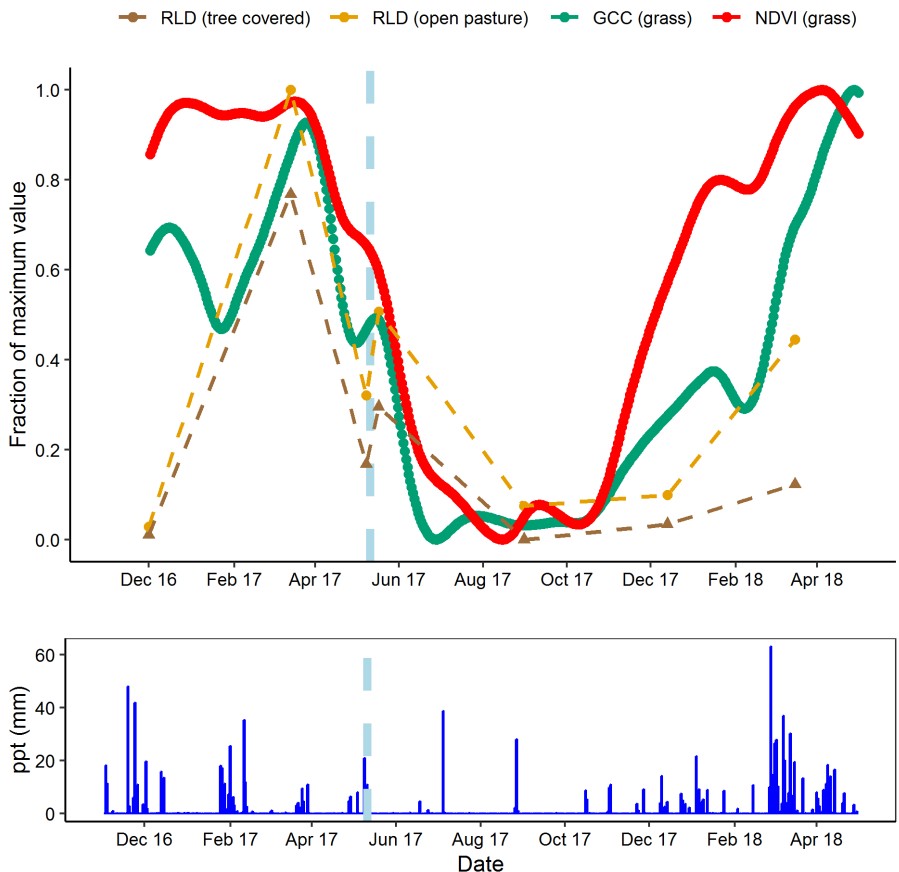

**Figure 8.** Comparison of minirhizotron root length density (RLD, mm mm$^{-2}$) dynamics at 9 cm depth for open pasture with site-level grassland. Normalized Difference Vegetation Index (NDVI) and grassland Green Chromatic Coordinate (GCC). After the rain pulse in May 2017 (indicated on graph, light blue dashes), minirhizotron measurements could detect the similar response to above-ground (shown in more detail figure 7). Desynchronization was evident in both autumn periods where proximal-remote sensed metrics reached near-peak levels while RLD remained low.

**Table 1.** Summary of pretreatment measures within EC footprints ('treatments'). All measurements were made in Nov 2013 for soil and in spring 2014 for plants. All soil properties are measured on surface (0-5 cm) soil and open pasture leaf N:P are a biomass weighted aggregate of samples of grass and forbs ( 98 % of herbaceous layer biomass). All errors are standard deviations. Soil nutrient contents (C,N,and Olsen-P) were always significantly higher in tree covered than open pasture microhabitats ($P < 0.005$, $P < 0.005$, $P < 0.05$ respectively). There were no significant effects due to footprint except soil pH (*, $p < 0.05$) but this did not result in a difference of weighted mean vegetative N:P ratios, so we considered this unimportant for the overall design.

| Treatment | Control | +N | +NP | Control | +N | +NP |
|---|---|---|---|---|---|---|
| Habitat | Open Pasture | | | Tree Covered | | |
| Soil Texture (Clay:Silt:Sand) | 6:20:74 | 5:20:75 | 6:21:73 | 1:20:79 | 5:20:75 | 8:25:67 |
| C (mg g$^{-1}$) | 4.87 ± 1.4 | 5.33 ± 1.2 | 6.56 ± 1.6 | 13.59 ± 0.5 | 10.06 ± 1.2 | 12.57 ± 3.8 |
| N (mg g$^{-1}$) | 0.86 ± 0.1 | 0.93 ± 0.1 | 1.04 ± 0.1 | 1.54 ± 0.1 | 1.30 ± 0.2 | 1.49 ± 0.4 |
| Soil C:N | 5.7 : 1 | 5.7 : 1 | 6.3 : 1 | 8.9 : 1 | 7.8 : 1 | 8.4 : 1 |
| Olsen-Extract P ($\mu$g g$^{-1}$) | 2.3 ± 0.6 | 3.68 ±1.4 | 3.38 ± 2.0 | 5.44 ± 0.9 | 5.45 ± 3.8 | 3.77 ± 1.4 |
| Soil pH | 5.42 ± 0.1 | 5.56 ± 0.5 | 4.93 ± 0.3 * | 5.50 ± 0.6 | 5.58 ± 0.4 | 4.93 ± 0.3 * |

| Treatment | Control | +N | +NP |
|---|---|---|---|
| herbaceous Layer Leaf N:P | 13:1 | 13:1 | 13:1 |

**Table 2.** Absolute (Herbaceous layer) Biomass Measurements and Root:Shoot Ratios at two points in 2017. Root shoot ratio increased later in the growing season but tended to decrease in nutrient added treatments, with the exception of N:pasture in March and N:tree covered in May which exhibited unusually high root:shoot ratio.

**March 2017**

| Nutrient | Vegetation | Aboveground Biomass | Belowground Biomass | Root:Shoot Ratio |
| --- | --- | --- | --- | --- |
| | kg ha$^{-1}$ | kg ha$^{-1}$ | | |
| Control | tree covered | 290 ± 20 | 5770 ± 320 | 20 ± 2 : 1 |
| Control | pasture | 290 ± 20 | 4510 ± 250 | 15 ± 1 : 1 |
| N | tree covered | 340 ± 20 | 6890 ± 390 | 21 ± 2 : 1 |
| N | pasture | 290 ± 20 | 7590 ± 420 | 26 ± 3 : 1 |
| NP | tree covered | 380 ± 30 | 6380 ± 350 | 17 ± 2 : 1 |
| NP | pasture | 300 ± 30 | 4900 ± 270 | 16 ± 2 : 1 |

**May 2017**

| Nutrient | Vegetation | Aboveground Biomass | Belowground Biomass | Root:Shoot Ratio |
| --- | --- | --- | --- | --- |
| Control | tree covered | 260 ± 40 | 6900 ± 700 | 26 ± 5 : 1 |
| Control | pasture | 370 ± 60 | 4280 ± 330 | 12 ± 2 : 1 |
| N | tree covered | 220 ± 30 | 8250 ± 640 | 38 ± 6 : 1 |
| N | pasture | 400 ± 50 | 5960 ± 580 | 15 ± 2 : 1 |
| NP | tree covered | 360 ± 40 | 7850 ± 900 | 22 ± 3 : 1 |
| NP | pasture | 440 ± 30 | 6670 ± 700 | 15 ± 2 : 1 |