# Peer review of "N:P STOICHIOMETRY AND HABITAT EFFECTS ON MEDITERRANEAN SAVANNA SEASONAL ROOT DYNAMICS"

_Biogeosciences, 2018_

## Referee Comment (RC1) · Anonymous Referee #1 · 19 Oct 2018

I think this manuscript presents an interesting study setting that is not often seen, but the manuscript cannot be published in its current state because of inadequate English expression and somewhat logical consistency. Huge amounts of data were obtained from your experiment, but the whole "story" was not clear enough from the data you have shown, you may delete some of the data to make the expression more concise and stronger. The English and details of expression, such as page 1 Line 20 is carbon (C) presented here the same with land C sink seen in line 2, if so, the first appearance should be abbreviated. Line 22 "site" or place? Line 26 what does it mean by "both global change factors", it should be clearer, etc. must be improved before it can be considered for publication. I also cannot imagine the experimental set up, a figure

would be very useful. Besides, in nutrient treatments (N and NP addition), why there was no P addition and how can you tell the relative importance or effects of N or P addition on your evaluated parameters? It needs more information. Therefore, I recommend reject with resubmission encouraged.

---

## Author Comment (AC1) · 27 Oct 2018

We thank the anonymous reviewer for their comments on our manuscript. We are pleased the reviewer recognized the novelty of the study setting and the amount of data collected. We are definitely willing to improve the clarity by adding a map of the field site, and try to streamline some expressions and adjust the short number of minor corrections requested in the first two paragraphs of the manuscript.

However, it is hard to understand the reviewer's statement regarding the clarity of the manuscript, because he/she hasn't described any major failures in the text in the short review. Particularly, we would grateful if the reviewer could accompany the recommendation to delete some of the data recommend any particular part of the dataset which they believe superfluous to the overall message because at the current stage is hard to understand what are the concerns of the reviewer which lead to their suggestion for an overall restructuring of the manuscript.

We will improve the description in the experimental design, which can all be found in the same paragraph at the beginning of the methods section (page 4 line 20 – page 5 line 1). We are also happy to provide an additional figure or table in the revised version to clarify the set up, which consists of 3 large scale ($\sim$ 20 Ha) nutrient treatments ( page 4 line 20-26), an installation of root observatories within each nutrient treatment (12 observatories per treatment, line 30), with observatories arranged in sets of 4 around individual trees (page 4, line 31), so each 'tree location contains two 'tree covered' and two 'open pasture' locations (page 5, line 1) per tree. This results in 6 replicates for nutrient treatment x location per footprint. To make sure this section is fully understandable we will add a clarification into the text on page 4, line 23, and define 'treatment' is as used to refer to nutrient treatments and tower footprints while 'location' refers to spatially distinct tree covered or pasture locations.

We would also like to provide a full response to the reviewer's concern that "in nutrient treatments (N and NP addition), why there was no P addition and how can you tell the relative importance or effects of N or P addition on your evaluated parameters? It needs more information"

We discuss the expected effects of nutrient treatments in both the introduction and discussion of the manuscript (e.g. page 4, lines 1-5, page 10, lines 22 to 25), which follows ecological theory regarding stoichometric imbalances induced by N available [1,2] rather than testing the effects of P addition. We make no claims at any point in the manuscript to be testing the effect of P availability alone on the parameters evaluated in the experiment but we will be happy to clarify some aspects (including some of the information which follows) if unclear.

The overall (site level ∼ 24 Ha) experiment was designed to study the impact of the stoichiometric N/P imbalance (high N/P in vegetation pools ratio (+N) vs control (control) level vs relieved limitation (+NP)) at ecosystem scale on ecosystem fluxes, functioning and to plant and soil processes. For this purpose a P treatment was not necessary. However in previous publications we have shown a small scale manipulation experiment using a full factorial design to evaluate the relative importance of P but with less frequent measurements and no minirhizotron data. In gross primary productivity, ecosystem respiration, and vegetation structure and vegetation structure (LAI, Chlorophyll content derived from proximal sensing information [3,4], and the biological turnover of P and soil Carbon, total Nitrogen and P-Olsen content as consequence of the nutrient status induced by the fertilization [5]). All these analyses show that the addition of P alone does not significantly impact vegetation processes, and structure, and while this did increase P in leaf tissues, this significant change observed is an increase in the P turnover rate and an increase of P in the leaf tissues, this did not significantly affect productivity, vegetation structure and carbon-water fluxes measured. Moreover, the vegetation pools at the site prior to fertilization indicate that the site is more N limited than P limited given the relatively low N/P ratio in plant tissues (N/P = 9.018 (sd 0.49)) measured in March 2014 [4]), and this explains the small response of ecosystem function, structure and function to P fertilization.

Regarding the specific points raised by the reviewer, which all concern the first two paragraphs of the manuscript, we agree that:

i) C as an abbreviation for carbon should be presented at first mention (page 1, line 2) rather than on page 1, line 20

ii) there could be some minor confusion with the technically incorrect use of 'both' to refer to the following list of global change factors (page 1, line 26), as this by strict definition refers to a list of two items only. This can be replaced with a small grammatical change in a further revision.

We however disagree that there is any confusion in using site to refer to a location (page 1, line 23) - when used as a noun, site always refers to a location and has essentially the same meaning as 'place'.

---

## Author Comment (AC2) · 8 Nov 2018

We have realized that our previous comment omitted the references, which are presented here:

References: 1. Peñuelas J, Sardans J, Rivas-ubach A, Janssens I a. 2012 The human-induced imbalance between C, N and P in Earth's life system. Glob. Chang. Biol. 18, 3–6. (doi:10.1111/j.1365-2486.2011.02568.x) 2. Peñuelas J et al. 2013 Human-induced nitrogen–phosphorus imbalances alter natural and managed ecosystems across the globe. Nat. Commun. 4. (doi:10.1038/ncomms3934) 3. Migliavacca M et al. 2017 Plant functional traits and canopy structure control the relationship be-

tween photosynthetic CO2uptake and far-red sun-induced fluorescence in a Mediterranean grassland under different nutrient availability. New Phytol. 214, 1078–1091. (doi:10.1111/nph.14437) 4. Perez-Priego O et al. 2015 Sun-induced chlorophyll fluorescence and photochemical reflectance index improve remote-sensing gross primary production estimates under varying nutrient availability in a typical Mediterranean savanna ecosystem. Biogeosciences 12, 6351–6367. (doi:10.5194/bg-12-6351-2015) 5. Weiner T, Gross A, Moreno G, Migliavacca M, Schrumpf M. 2018 Following the turnover of soil bioavailable phosphate in Mediterranean Savanna by oxygen stable isotopes. (doi:10.1029/2017JG004086)

---

## Referee Comment (RC2) · Anonymous Referee #2 · 9 Jan 2019

Studies on root systems are extremely labor intensive and thus data on fine root growth dynamics are very limited, especially in Mediterranean systems. The manuscripts presents a thorough assessment of fine root dynamics in a Mediterranean savanna and shows that above and belowground growth are coupled and respond sensitive to seasonal moisture changes. The data clearly contributes to our knowledge on the belowground system, but in its current state the manuscript is strongly limited by (1) the poor experimental set-up with unrepeated fertilization treatments whose impact on ecosystem stoichiometry remains uncertain, (2) an unclear and puzzling presentation of data. (3) Including auxiliary data (N-P contents, soil moisture, soil properties) would greatly improve the quality and the interpretability of the manuscript and in such a large

scale study, these data should also be available. (4) Finally, the language is somewhat sloppy, see many examples in the Specific comments. Based on these points, the manuscript does not meet the standards of Biogeoscineces and requires a major input of additional data, a rewriting of large section and thus should be submitted as a new manuscript.

(1) The overall experimental set-up is not ideal for a root study as there are no real replicates for the treatments. I can understand that the sampling design has been adjusted to the overall design of the eddy-covariance measurements (not allowing more replications), but the manuscript profits very little from these additional measurements. I would recommend to state this statistical flaw openly in the Statistic section (and readers can judge by themselves if they accept the design as pseudo-replication or not. Moreover, site and soil of the different sites have to be presented if they are comparable among sites. Also, by analyzing N-P contents in fine roots or by presenting other indices of N-P fertility, the authors could provide evidence that the differences among sites can be indeed attributed to fertilization. In the Discussion, it is written that "plants exploring shallow soils in the P-limited N treatment" – which actually indicates that soil conditions were apparently not similar among treatments or sites – this definitely has to be documented! P fertilization is particularly prone to become sorbed in soils and thus without additional documentation of the effectiveness of the treatment, the discussion of P effect or P deficiency remains speculative. In this sense, also the title is misleading as the authors did not study stoichiometry but fertilization.

(2) The data presentation remains unclear as it is extremely difficult to follow what has been measured at what time as the description in the method section differs from what is shown in the Figures and Tables. Figures 2, 4 and Table 1 only show subsets of the data differing among themselves and it remains unknown why. The same holds true for the soil core and ingrowth measurements, where I presume that the labelling in Figure 4 was simply mixed up. (3) The manuscript would greatly profit from auxiliary data. Indices of N-P-fertility (ideally in root tissues) to document if treatment effects are

real (or simply related to site differences), soil properties to document micro site effects (highlighted in the Introduction), and seasonal changes soil moisture contents (to discuss habitat effects and potentially also critical moisture contents). Specific comments: (1) Data presentation. Figure 2 shows data from March 17, May 17, Sep 17, Dec 17 while in the methods it is written that RLD has been measured on six dates (as shown in Figure 6). Figure 4 shows other dates. Why are the data from the other dates not shown?

Comments to Figures:

Figure 4 start y –axis at 0!

First sentence of Figure Caption: Root biomass in top 13 cm from ingrowth cores (all installed Dec. 2016 and removed then in Nov 16 and Mar 17): how can you install a core in Dec 2016 and sample it in Nov 2016, hence a month earlier? The method text on page 6 reads: We collected ancillary measurements of root biomass from two non-concurrent methods, sampling within 1m of the minirhizotron tubes; soil cores (Dec 2016, March 2017) and two rounds of ingrowth cores installed in December 2016 (removed May 2017, December 2017) and December 2017 (removed March 2018, May 2018).

The color code on Figure 4 indicates 2 for ingrowth cores (Nov 16 and Mar 17 on the left) and 4 sampling dates for soil cores – I presume that these have been mixed up

Figure 5 Caption: Control Treatment response to rain pulse in May 2017 (Section 3.5). Response of??? Please keep correct and self-explanatory in Figure captions. As a reader I would also prefer if you would avoid abbreviations here

Figure 6: What is the unit of y-axis: % of maximal value, only 1%. Probably you mean 'fraction'. Explain the abbreviations please

Table 1: Why don't you report the data from the September and December 2017 measurements?

Comments to text

Introduction

Page 2 : 12-15 sentence read awkward

Page 2 L. 17 'ecological attention' – please clarify

Page 2 L. 25ff is a listing of global change buzzwords that are not linked to this study – instead the authors should develop how N and N-P additions may affect root growth. This appears not before introducing the experimental set-up and phrasing the hypothesis.

P. 3L.10 rephrase 'a 2015 meta analysis'

P. 3 L. 14, L. 22 'fewer investigating global change factors' - buzzwording, which one did the authors? The specific one of this study needs to be developed

P. 4 L. 2 'standard resource limitation theory' - explain what standard is

Results:

P. 7L. 22 cover-based markup is slang

P 8 L. 3 ff in the most abundant period – reads sloppy to me

P. 8 L. 4 peak shallow RLD – please explain

P. 8 L. 5 "shallow soil" - which soils were shallow? – this needs to be described in the methods

P. 8 L. 9 less roots in TC than in OP – what is the statistical significance? While the experimental design does not allow an analysis of treatment effects it allows the identification of tree canopy effects.

P. 8 L. 17 "limited the potential for variation" please explain – variation should be independent from intensity

P. 8 L. 20 "While these" clarify what "these" are!

P. 8 L.25 "as following transformation to fit the assumption of linear models" – requires explanation

P.9 paragraph 3.5. While a higher temporal resolution of root growth is highly appreciated, 2 time points are still providing a poor picture and the reader is left how significant the findings are.

P. 9 L. 16 sentence is awkward and unclear – "indicate" is somewhat strong

P. 9 L. 20 introduce RSR, try to avoid a abbreviations, the vegetation layer meant is grassland or?

P. 9 L. 22: 3-4 statements in one sentence is too much

P. 9 L. 23/24 statement is unclear. Do you really want to compare the differences between roots and shoots as compared to nutrient treatment effects?

Discussion

Overall, the Discussion is somewhat lengthy partly due to the repeating of methodological issues.

P. 9 L. 27 clarify what the different directional effect

P. 10 L. informed by installation and sampling effort: clarify what is meant by "informed"

P. 10 what is an "Effect on root measurements between methods" Provide a proper title of this paragraph!

P. 11 L.1 "inversion of directional effect" – requires explanation

P. 11 L. 8 the sentence on NP effect interrupts the discussion on habitat effects

P. 11 L. 11-17 methodological issues have already been discussed in the first paragraph of the discussion

P. 11 L. 25 plants exploring shallow soils in the P-limited N treatment – here we learn that soil conditions were apparently not similar among treatments or sites – this has to be documented

P. 11 L. 27 the discussion on P availability effects has to be backed up with some measures of P availability

P. 12 L. 2 "elevation of N and NP biomass – slang and unclear if above- or belowground biomass - explain!

P. 12 L.10 the discussion on plant trait effects could be more straightforward by simply presenting species composition of the grassland in your study

P. 12 L. 15 "under P deficiency" – speculation

P. 12, L. 29 " so this relative decrease in RSR was still a large absolute increase in root biomass" – needs further explanation – is it not evident that a ratio changes with its numerators and denominators. . .

P. 13L 5 what are the sampling depths of the other studies, frequently they are limited to surface soil, but the data of this study indicate also that roots are present in the deeper soil.

P. 14 L. 1 "C status" requires explanation or omit

P. 14 L. 7 "short-term events are highly coupled" – awkward wording you probably mean above and belowground growth in response to an event and not the event as such. . ..

---

## Author Comment (AC3) · 8 Feb 2019

We are grateful to anonymous reviewer 2 for the care and attention they have put to this detailed review of the manuscript. We agree with the majority of their concerns, which to us appear to be addressable with the addition of pre-existing data to the manuscript and changes in the text of the manuscript to improve the clarity. Here we would like to respond in detail to the main points raised in the review, which we will show before each response in **bold**.

**1 Main Comments**

**"The overall experimental set-up is not ideal for a root study as there are no real replicates for the treatments. I can understand that the sampling design has been adjusted to the overall design of the eddy-covariance measurements (not allowing more replications), but the manuscript profits very little from these additional measurements. I would recommend to state this statistical flaw openly in the Statistic section (and readers can judge by themselves if they accept the design as pseudo-replication or not"**

We acknowledge that the experimental design should be better explained in the manuscript and would like to add a paragraph similar to the following in the Methods section:

*Studying whole ecosystem responses of gas exchange to experimental manipulation requires the eddy covariance technique. Given the large area (in our case, 20 hectares) covered by these measurements, which covers much small scale variation, and for logistical reasons of finding large enough areas with homogenous land cover, these measurements are typically not spatially replicated. Ecosystem properties and the results of statistical tests of pre-treatment measures (including soil N and P, leaf level nutrients and CO2 fluxes, as detailed below) indicate that variation was greater within than between treatment footprints before application of fertilizers. On the other hand, root analyses, as detailed in this paper, are point measurements which have to be replicated in space to cover site-level variation within treatments. Replicates of root analyses per treatment as conducted in the presented study inform about the variation within each footprint area and are therefore pseudo-replicates. This violation of independence was a necessary constraint of the experiment given the wildly different spatial scales employed at our field site and ambitions for a larger scale synthesis of data not presented in this study. Individual sampling trees were randomly located from*

*around 480 trees per 20 ha treated plot (with a few constraints based on spacing of trees to ensure grassland for 'grassland' microsites). The median distance between individual pairs of trees and open pasture locations within a treatment is 116 metres (60 meters minimum). For the statistics employed in this study, we assume for all root analyses that individual minirhizotrons and associated root measurements are repeated measures of the same individual 'sample' and that 'samples' are nested within one of three individual trees within treatment. Due to the lower number of replicates in soil data (in the rest of this response), we removed this nested term in order to avoid overfitting the statistical models employed. However, individual minirhizotrons were always at least 5 m apart, and arranged linearly, so we expected this spatial co-variation to be minimal, especially between the grassland locations (on opposite sides of each tree location) where the biggest treatment-level differences in soil tended to be found.*

In addition, we would like to make an addition to the methods section of Figure 1. (clarifying the set up in response to the first reviewer's comments, which should help with interpreting the experiment), and explaining in depth the pre-treatment similarity between the treatments:

**"Moreover, site and soil of the different sites have to be presented if they are comparable among sites. Also, by analyzing N-P contents in fine roots or by presenting other indices of N-P fertility, the authors could provide evidence that the differences among sites can be indeed attributed to fertilization."**

Root data was only available during the period of the experiment which is the main focus of this study. However, several other measures of nutrient contents and calculated stoichiometric ratios of plants and soil in the tower footprints indicated that the plant-soil system were similar before fertilization and changed in the manner assumed by the discussion in our manuscript after fertilization. Hence we will answer this comment in two parts: 1) demonstrating pretreatment similarity, and then 2) post-treatment differences in measures of N-P fertility. These data (excepting where cited) are currently unpublished and we are happy to include some version of this in a revision of

|  | Control | N | NP |
|---|---|---|---|
|  | Clay:Silt:Sand | Clay:Silt:Sand | Clay:Silt:Sand |
| Open Pasture (OP) | 12:24:64 | 11:25:64 | 9:23:68 |
| Under Canopy (UC) | 8:26:65 | 10:14:76 | 10:18:72 |

**Table 1.** Table 1: Comparison of soil texture between tower locations. All sites were comparable with a tendnacy for slightly more Clay and Silt in pasture locations. Variance (not shown) was high and there was no significant difference between treatment nor habitat.

the manuscript or as appendices alongside a slightly expanded method section with full details of the measurement protocol. In all statistical tests shown below, we either use a mixed effects model formulated as in the statistical tests in the manuscript, or an ANOVA followed by Tukey Honest Significant Difference (Tukey HSD) for multiple comparisons between treatments within habitat and campaign.

1) Pretreatment Similarlity between Treatments

We performed the experiments detailed in this study from 2015 to 2018 after the application of nutrients in two of the three EC footprints. On the ecosystem scale, the three tower footprints were comparable pre-fertilization in a variety of stand characteristics such as the fraction of grass and tree cover, as well as C fluxes and footprint characteristics (Figure 2, El-Madany et al., 2018)

For the soil, the plots were also similar pretreatment (Nov 2013). Soil texture changed with depth but did not vary between plots. There were no significant differences in soil texture between sites. Soil texture from 0-60 cm was slightly sandier (9

In the pretreatment surface (0-5 cm) soil, C (control plot: 7.46 $\pm$ 4.0, N plot: 6.62 $\pm$2.6, NP plot: 7.28 $\pm$ 2.6 g kg-1 $\pm$ sd. ) and N (control plot: 0.73 $\pm$ 0.3, N plot :0.73, $\pm$ 0.2, NP plot: 0.69 $\pm$ 0.2 g kg-1 $\pm$ sd. ) did not differ between treatments with a C:N ratio of about 10:1 across all treatments. The C and N soil sampling made at this time did not sample in areas specifically under trees (unlike our design), but spatial variation is

well documented in Dehesa landscapes, with higher N contents under trees (Gallardo, 2003; Gallardo et al., 2000) than in pastureland. Separate measurements made pre-treatment (also November 2013) measured phosphorus with the Olsen method. Soil phosphorus (Olsen method) in surface soil was around 4 ug g-1 (data shown in figure 5). There were no significant differences (P > 0.05) in this measure between individual tower footprints nor habitats.

Likewise, there were no significant differences (P > 0.05, data shown in response to specific 'stoichiometry' comment) in N:P ratio of pasture plants between the tower foot-prints in March 2014 ( one year pre – P treatment, shown in figure 8). In May 2014 (also pretreatment), there was a small significant difference in the NP treatment for this ratio. As this was in the opposite direction from later treatment effects (indicating more N compared to P in the pretreatment NP treatment) we consider this as possibly a type I error at this time and unimportant for the overall experimental design.

2) Post-Treatment Differences in N:P fertility

Immediately after fertilization, bioavailable (Resin-P) was significantly higher in the NP footprint than the control and N footprint (Weiner et al., 2018). Extractable soil N was not measured at these times but was at subsequent time points during the experiment, during the majority of the root sampling campaigns.

During the period of the root experiment detailed in this paper, we made measurements of 'plant available' extractable soil P using the Olsen method, total N contents and soil inorganic N extractions (2M KCL) in 0-5 cm surface soil at 2/3 of the microsites used in this study. Total N content (Figure 3) was fairly consistent through time but significantly affected by Habitat (P< 0.001) and the N treatment (P < 0.01) but not the NP treatment (P > 0.05). In general, N contents in UC locations were both higher and more variable than in OP locations where N contents were lower but with a clearer treatment effect.

During the 2017-2018 period of study, inorganic N availability as measured by KCL extraction on surface soil were 2.37 ± 3.8 (sd.) ug g-1 in UC locations and 1.79 ± 3.1

ug g-1in OP locations (Figure 4). Overall there was a significant effect of habitat (P < 0.05) on extractable inorganic N and a borderline non-significant (P = 0.07) effect of N treatment. This was driven by very high N contents (up to 22 ug g-1) in some samples from the N treatment in March 2017 - later in the experiment these differences were much smaller and at individual dates there were not significant differences between treatments. Mineral N pools are driven by mineralization rates and biological uptake and for mineralization which so it is not clear if this declining difference and the difference to the NP treatment were due to changing N availability or depletion of this pool by uptake from plants with a greater access to available P.

For phosphorus, there was a significant of the NP treatment (P < 0.001) and habitat (P < 0.001) on Olsen-P content (Figure 5). In the control and N treatment, mean extractable P was 3.00 ± 2.78 ug g-1 in UC locations and 1.44 ± 0.9 ug g-1 in OP locations. When P was added in the NP treatment, these phosphate-P concentrations were 7.03 ± 5.6 ug g-1 in UC locations and 3.5 ± 1.54 g-1 in OP locations.

We also can show data on above-ground pasture plants responses (limited to sampling in the 'OP' areas, Figure 6.). N and P contents were fairly variable, but after fertilization both treatments diverged from the control with more leaf P in the NP treatment (P < 0.001 for NP only) and significantly elevated N both treatments (P < 0.001 for both) over the entire post-treatment period.

**"Also, by analyzing N-P contents in fine roots**..."

We agree that N-P contents from fine roots would be interesting to explore the effects of the fertilization. However, as we can demonstrate stoichiometric effects in surface soil, we do not think that this is essential to support the argument of differing site nutrient stoichiometry (which may, or may not be reflected in the plastic plant expression of elemental concentration of tissues) and its effect on root biomass and RLD over time and between treatments. We intend this manuscript to focus on root biomass dynamics and consider N:P content of roots (while available on a very limited scale within this

experiment and intended for publication in another study) not essential for the main message.

**"In the Discussion, it is written that "plants exploring shallow soils in the P-limited N treatment" – which actually indicates that soil conditions were apparently not similar among treatments or sites – this definitely has to be documented!"**

We did not intend to suggest that there was a difference in soil depths between individual microsites (at every location measured we were able to install a minirhizotron observatory up to 40 cm vertical depth, with no measurements below this depth), but rather shallow in the sense of distinguishing between the upper part of the soil profile and subsoils. In a revision of the manuscript we will rephrase this to remove any potential ambiguity, e.g. to 'a proliferation of roots in the topsoil'.

**"P fertilization is particularly prone to become sorbed in soils and thus without additional documentation of the effectiveness of the treatment, the discussion of P effect or P deficiency remains speculative. In this sense, also the title is misleading as the authors did not study stoichiometry but fertilization."**

We have included additional data in the preceding graphs which indicate a difference in P availability. In the next comment, we will show differences in stoichiometry.

**"In this sense, also the title is misleading as the authors did not study stoichiometry but fertilization."**

We agree that in order to support the assumptions made in the discussion section as well as the choice of title ('stoichometry' rather than 'fertilization'), additional data on site-level stoichiometry is useful. The ratio between these mineral forms of the ions (N:P) in the Olsen-P and KCL-N extracts from the 0-5 cm soil indicated a strong treatment effect (P < 0.001, Figure 7). The N treatment had a higher (P = 0.06) ratio and the NP treatment a lower (P < 0.001) ratio than control. This difference was stronger

than the habitat effect (P = 0.08) in the experiment on this ratio.

While we do not have a long-term root N:P timeseries following fertilization, the pasture plant leaf N:P ratio (equivalent to OP locations, Figure 8) diverged from control for the N treatment only (Control-N P < 0.001, Control-N P > 0.05). These data were quite variable, especially shortly after treatment. Changes in the NP treatment were more complex (sometimes with higher N:P ratios than control, sometimes equal or lower), probably due to seasonal changes in diversity, or plastic tissue responses of an aggregate of multiple species.

**"The data presentation remains unclear as it is extremely difficult to follow what has been measured at what time as the description in the method section differs from what is shown in the Figures and Tables. Figures 2, 4 and Table 1 only show subsets of the data differing among themselves and it remains unknown why. The same holds true for the soil core and ingrowth measurements, where I presume that the labelling in Figure 4 was simply mixed up**. . . . **Figure 2 shows data from March 17, May 17, Sep 17, Dec 17 while in the methods it is written that RLD has been measured on six dates (as shown in Figure 6). Figure 4 shows other dates. Why are the data from the other dates not shown?"**

We are very grateful for the reviewer's thorough identification of several mistakes in the figures presented in the original submission. Some of the confusion about measurements may have also arisen from incorrectly stating that there were six minirhizotron campaigns in the original methods when in fact there were seven (as evident from Figure 3). The mistakes identified have been corrected as the reviewer suggested (detailed in specific comments) and may address some of this concern. However we also can address the difference in each figure in detail:

Figure 2 shows only dates in 2017. We intended this subset to show clearly the change in vertical profile, while Figure 3 shows the change over time at selected depths.

Figure 4 shows root biomass from two methods over six sampling campaigns, as indicated in the methods section (soil cores in Dec 2016 and March 2017, ingrowth cores collected in May 2017, December 2017, March 2018, and May 2018). Physical root measurements were not collected at every minirhizotron sampling date. This was due to logistical constraints and physical difficulty in sampling soils over the dry summer. Therefore these data are not available for each campaign.

As indicated in the legend, Table 1 shows the spring growing period data over both years for the purpose of comparison. During the growing season, biomass was larger and treatment means diverged more than during fallow periods and so these were included alone to emphasise differences in this period of the year. We also erroneously formatted this table during LaTEX conversion and one of the column headings was lost.

In cases where we have omitted data for the sake of clarity of the figures, we are happy to include it if the reviewers do not agree with the logic of our selections.

**"The manuscript would greatly profit from auxiliary data. Indices of N-P-fertility (ideally in root tissues) to document if treatment effects are real (or simply related to site differences), soil properties to document micro site effects (highlighted in the Introduction)"**

We have shown graphs of differing N:P fertility between treatments above and explained why we do not show root tissues. The above-ground vegetation data clearly show a treatment effect.

**"The manuscript would greatly profit from** ... **seasonal changes soil moisture contents (to discuss habitat effects and potentially also critical moisture contents)"**

We have data on seasonal changes in soil moisture contents at the treatment level and separated for habitat (UT and OP). However as this is not paired directly to minirhizotron / soil sampling we feel this is of limited use to fully discuss changes in root biomass and potential differences between microsites. We can include these full time

profiles of soil water content from the eddy tower footprints in supplementary material to this manuscript.

**2 Comments - Figures**

**Figure 4 start y –axis at 0!**

The y axis of all figures included in the manuscript either starts at 0, or is reversed (in the case of vertical profiles) with 0 at the top.

**Figure 4 caption - mistake in text**

This is correct, there is a mistake in the text. This should read in accordance with the methods and will be corrected.

**The color code on Figure 4 indicates 2 for ingrowth cores (Nov 16 and Mar 17 on the left) and 4 sampling dates for soil cores – I presume that these have been mixed up**

This is correct and can be easily corrected.

**"Figure 5 Caption: Control Treatment response to rain pulse in May 2017 (Section 3.5).Response of???  Please keep correct and self-explanatory in Figure captions. As a reader I would also prefer if you would avoid abbreviations here"**

This caption should read ' Control Treatment root length density (RLD) response to rain pulse in May 2017 (Section 3.5). Repeat minirhizotron sampling shows increases in RLD throughout the soil profile ($P < 0.001$ in for all locations)'

**"Figure 6: What is the unit of y-axis: % of maximal value, only 1%. Probably you mean 'fraction'. Explain the abbreviations please"**

This axis title should read 'Fraction of Maximal Value'. We will add units to the figure

legend and define RLD in the caption for this figure.

**"Table 1: Why don't you report the data from the September and December 2017 measurements?"**

We have already responded to this above in the section beginning with the comment 'The data presentation remains unclear. . .'

**3 Specific Comments**

**Page 2 : 12-15 sentence read awkward**

We have altered this sentence in the manuscript

**Page 2 L. 17 'ecological attention' – please clarify**

We have rephrased this to 'Plant phenology studies'

**Page 2 L. 25ff is a listing of global change buzzwords that are not linked to this study – instead the authors should develop how N and N-P additions may affect root growth.This appears not before introducing the experimental set-up and phrasing the hypothesis.**

We acknowledge that this part of the manuscript can be improved by moving the explanation of how the N and N-P additions can affect root growth to replace this section.

**P. 3L.10 rephrase 'a 2015 meta analysis'**

We are reluctant to rephrase this sentence. The reference does indeed refer to a meta-analysis of root and shoot phenology and understanding the methodology of the study (and relative lack of Mediterranean system data) is relevant to discussing its results.

**P. 3 L. 14, L. 22 'fewer investigating global change factors' - buzzwording, which one did the authors? The specific one of this study needs to be developed**

The reviewer's request that we spend more time here to develop the specific question of this study is reasonable and can be easily addressed. We will replace this paragraph with one focusing specifically on nutrient and other limitations in Mediterranean systems such as the following: *In many cases, root growth is desynchronized from production of shoots (Blume-Werry et al., 2017; McCormack et al., 2014; Steinaker and Wilson, 2008) and linkages of root function and root dynamics are often poorly understood. As a major function of roots is nutrient uptake, supplying resources which are often limiting, nutrient availability may play an important role in regulating the timing and magnitude of root production. The role of nutrients on root production, which both are important themselves and may regulate responses to other factors, is particularly complex in Mediterranean grasslands (Dukes et al., 2005), where N inputs typically produce a biomass response in both shoots and roots, although belowground responses are less consistent than above-ground. This may relate to site-specific limitations and the balance between co-limitation of nitrogen and phosphorus, and water, all acquired by roots, and with changing availabilities throughout the year. In such seasonally arid and meditteraneans systems, co-limitations by both N and P are often suggested (e.g. Ries and Shugart, 2008; Sardans et al., 2012; Sardans and Peñuelas, 2013) although there is considerably less information on root responses which may respond in different ways to shoots, particularly under drought (Gargallo-Garriga et al., 2014).*

**P. 4 L. 2 'standard resource limitation theory' - explain what standard is**

The concept that plants invest resources to acquire their most limiting resource (e.g. C, nutrients, water) is well established in ecology and elaborated on in the following lines in the manuscript. We will expand this section to better explain the theory in revision to reduce any ambiguity.

**P. 7L. 22 cover-based markup is slang**

We agree that this can be phrased more specifically and is changed in the revised manuscript to 'our markup method'

**P 8 L. 3 ff in the most abundant period – reads sloppy to me**

We have altered this to 'Period with the highest RLD'

**P. 8 L. 4 peak shallow RLD – please explain P. 8 L. 5 "shallow soil" - which soils were shallow?**

We have already elaborated over the unitentional ambiguity over our use of the word 'shallow' and will change this in the revision to make clear we are referring to locations closer to the soil surface.

**P. 8 L. 9 less roots in TC than in OP – what is the statistical significance? While the experimental design does not allow an analysis of treatment effects it allows the identification of tree canopy effects.**

This was an omission and we can confirm that habitat effects were highly significant although a difference between the two locations was expected from previous studies and not particularly interesting in the context of this study which included the tree and grass locations to incorporate this spatial difference into the nutrient experiment. We will revise this sentence to 'In general, the minirhizotron images contained less roots in TC than OP locations (Across the whole dataset, P < 0.001, Figure 3).

**P. 8 L. 17 "limited the potential for variation" please explain – variation should be independent from intensity**

This comment is correct and our point here should be rephrased to 'This difference tended to be greatest during the spring as low average RLDs outside the main growing season meant absolute differences between treatments, if they existed, were impossible to detect using our methodology'.

**P. 8 L. 20 "While these" clarify what "these" are!**

We have altered this to 'While both these measurements'

**P. 8 L.25 "as following transformation to fit the assumption of linear models" –**

**requires explanation**

We can add a brief elaboration on this point. In this comparison between RCI and RLD, data were transformed for normality. In the later mixed effects models the outcome was not transformed as this is not an assumption, but residuals were checked for normal distributions.

**P.9 paragraph 3.5. While a higher temporal resolution of root growth is highly appreciated, 2 time points are still providing a poor picture and the reader is left how significant the findings are.**

We agree with the reviewer here (which points to a need for higher time resolution root measurements) but do not think there is a possible change to the manuscript that improves this

**P. 9 L. 16 sentence is awkward and unclear – "indicate" is somewhat strong**

We have altered 'indicate' to 'suggest'

**P. 9 L. 20 introduce RSR, try to avoid a abbreviations, the vegetation layer meant is grassland or?**

RSR was introduced in P6 L27 but in order to reduce this potential confusion we will remove this abbreviation and replace it with 'root: shoot ratio' throughout the manuscript This should read 'herbaceous layer' in order to exclude above ground biomass including trees.

**P. 9 L. 22: 3-4 statements in one sentence is too much**

We agree and will split this sentence

**P. 9 L. 23/24 statement is unclear. Do you really want to compare the differences between roots and shoots as compared to nutrient treatment effects?**

The purpose of this statement is to point out that the effect of this experiment is on

absolute root biomass, rather than a shift in above-belowground allometry

**Overall, the Discussion is somewhat lengthy partly due to the repeating of methodological issues.**

We will remove this repetition in revision

**P. 9 L. 27 clarify what the different directional effect**

This sentence in the manuscript is incorrect * We will replace this with 'Notably, we found that the effect of habitat type reversed between RLD (measured from minirhizotrons) and root biomass (measured directly). Separating this potential interaction from effects arising from methodological differences is important to consider. '

**P. 10 L. informed by installation and sampling effort: clarify what is meant by "informed"**

We will delete the latter part of this sentence, from 'informed' onwards.

**P. 10 what is an "Effect on root measurements between methods" Provide a proper title of this paragraph!**

For clarity, we will change the title of this paragraph to 'Treatment and Habitat Effects on Root Length Density and Root Biomass'

**P. 11 L.1 "inversion of directional effect" – requires explanation**

We will replace this with 'This difference between above and below-ground biomass'.

**P. 11 L. 8 the sentence on NP effect interrupts the discussion on habitat effects**
We agree and will move this sentence

**P. 11 L. 11-17 methodological issues have already been discussed in the first paragraph of the discussion**

We agree and will move this brief discussion to the first paragraph of the discussion

**P. 11 L. 25 plants exploring shallow soils in the P-limited N treatment – here we learn that soil conditions were apparently not similar among treatments or sites – this has tobe documented**

This is a misinterpretation as already discussed. In addition, we have moved this sentence to the end of the paragraph in order to clarify the argument being made.

**P. 11 L. 27 the discussion on P availability effects has to be backed up with some measures of P availability**

This is addressed in the main comments

**P. 12 L. 2 "elevation of N and NP biomass – slang and unclear if above- or below-ground biomass - explain!**

We have replaced this with: 'In the soil cores, the increased biomass under both N and NP treatments may be indicative of overall changes in root architecture between treatments (including differing proportions of lateral roots) in the N treatment, which are not detected by our minirhizotrons.'

**P. 12 L.10 the discussion on plant trait effects could be more straightforward by simply presenting species composition of the grassland in your study**

A discussion on plant traits would indeed be more straightforward but we think a thorough analysis of the 'diversity effect' of the nutrient treatments in this study is beyond the scope of this paper. We considered the potential 'community average' trait effects reasonable to discuss given the different responses of minirhizotron-calculated RLD and absolute biomass observed in this experiment.

**P. 12 L. 15 "under P deficiency" – speculation**

This is addressed in the main comments

**P. 12, L. 29 " so this relative decrease in RSR was still a large absolute increase in root biomass" – needs further explanation – is it not evident that a ratio changes**

**with its numerators and denominators:**

We will replace this sentence with 'Compared to the other treatments, in the NP treatment root:shoot ratio decreased (as expected with decreased whole-plant investment into NP acquisition), despite the increasing root biomass.

**P. 13L 5 what are the sampling depths of the other studies, frequently they are limited to surface soil, but the data of this study indicate also that roots are present in the deeper soil.**

The other studies referenced in this paragraph sampled to 20, 100, and 150 cm. Generally herbaceous species in dehesa can have deep roots, although, as we show, the majority of roots are concentrated in the upper soil layers. Additionally, we have noticed a mistake in this paragraph, omitting one reference and an accidental substitution of a reference by a similarly named author. We will revise this in an updated version of the paper.

**P. 14 L. 1 "C status" requires explanation or omit**

we will omit 'C status' and remove the brackets

**P. 14 L. 7 "short-term events are highly coupled" – awkward wording you probably meanabove and belowground growth in response to an event and not the event as such:**

We will replace 'short term events' with 'short-term above- and below- ground plant dynamics'

**4 Full captions for figures**

These are the full captions for the figures included in the response, which were abbreviated by the webform.

Figure 1: Schematic diagram of experiment. In panel a) we show 12 minirhizotrons per treatment, split between three trees into 'OP' (open pasture) and 'UC' (under canopy) areas. This is a small subset of the total trees in each footprint. Median distance between trees within treatment is 116 metres and minimum distance between individual minirhizotrons is 5 metres. In panel b), we show conceptual release from N limitation in our N treatment but promotion of P limitation which is alleviated by NP additions.

Figure 2: Mean diurnal course of NEE (Net ecosystem exchange), LE (Latent heat), H (Sensible heat) and u* (Friction velocity) from March 25th to May 15th 2014. These data are from El-Madany et al., Agricultural and Forest Meteorology 262, 258-278, (2018), and are included here to support our argument, using slightly different nomenclature to this manuscript; CT is 'control tower', NT is 'nitrogen addition tower' and NPT is 'nitrogen and phosphorus addition tower'. This period is pre-treatment and the small differences between the towers in terms of these footprint-level measurements are much smaller than the uncertainties associated to the fluxes.

Figure 3: Total N in the 0-5 cm soil. This was higher in the N treatment ($P < 0.01$) and under canopy ($P<0.005$). Letters indicate TukeyHSD groupings of one-way ANOVA on habitat-date combinations.

Figure 4: 2M-KCL extractable N in surface soil. There were no significant differences between treatments within habitat and campaign but a significant effect of habitat over the whole period of the experiment ($P < 0.05$). Errorbars show standard error of the mean.

Figure 5: Olsen-extractable phosphate-P in surface soil. There were no significant differences before the treatment but during the period of the experiment, more P was found in the NP treatments and under canopy. Letters show Tukey HSD groupings within campaign-habitat combinations and errorbars show standard error of the mean.

Figure 6: Leaf N and P contents for pasture plants from 2014 to 2017. Over the entire period post-treatment, there is a significant elevated leaf P in the NP treatment ($P <$

0.001) and leaf N in both treatments (P < 0.001)

Figure 7: Ratio of extractable N (2M KCL) to extractable P (Olsen method) in 0-5 cm soil during the experiment. Over all measurements, the N treatment has a borderline significant (P = 0.06) greater N:P ratio and the P treatment a significantly lower (P < 0.001) ratio than control. Letters show Tukey HSD groupings within campaign-habitat combinations and errorbars show standard error of the mean.

Figure 8: Herbaceous Layer leaf N:P ratios form 2014 to 2017. There was a significant effect of both treatments on N:P ratio over the combined period post-treatment.

**5  References**

Blume-Werry, G., Jansson, R. and Milbau, A.: Root phenology unresponsive to earlier snowmelt despite advanced above-ground phenology in two subarctic plant communities, Functional Ecology, 31, 1493–1502, doi:10.1111/1365-2435.12853, 2017.

Dukes, J. S., Chiariello, N. R., Cleland, E. E., Moore, L. A., Rebecca Shaw, M., Thayer, S., Tobeck, T., Mooney, H. A. and Field, C. B.: Responses of grassland production to single and multiple global environmental changes, PLoS Biology, 3(10), e319, doi:10.1371/journal.pbio.0030319, 2005.

El-Madany, T. S., Reichstein, M., Perez-Priego, O., Carrara, A., Moreno, G., Pilar Martín, M., Pacheco-Labrador, J., Wohlfahrt, G., Nieto, H., Weber, U., Kolle, O., Luo, Y. P., Carvalhais, N. and Migliavacca, M.: Drivers of spatio-temporal variability of carbon dioxide and energy fluxes in a Mediterranean savanna ecosystem, Agricultural and Forest Meteorology, 262(July 2017), 258–278, doi:10.1016/j.agrformet.2018.07.010, 2018.

Gallardo, A.: Effect of tree canopy on the spatial distribution of soil nutrients in a Mediterranean Dehesa, Pedobiologia, 47(2), 117–125, doi:10.1078/0031-4056-00175,

2003.

Gallardo, A., Rodr, J. J., Covelo, F., Fern, R., Rodriguez-Saucedo, J. J. and Fernandez-Ales, R.: Soil nitrogen heterogeneity in a Dehesa ecosystem, Plant and Soil, 222(1–2), 71–82, doi:10.1023/A:1004725927358, 2000.

Gargallo-Garriga, A., Sardans, J., Pérez-Trujillo, M., Rivas-Ubach, A., Oravec, M., Vecerova, K., Urban, O., Jentsch, A., Kreyling, J., Beierkuhnlein, C., Parella, T. and Peñuelas, J.: Opposite metabolic responses of shoots and roots to drought, Scientific Reports, 4, 1–7, doi:10.1038/srep06829, 2014.

McCormack, M. L., Gaines, K. P., Pastore, M. and Eissenstat, D. M.: Early season root production in relation to leaf production among six diverse temperate tree species, Plant and Soil, 121–129, doi:10.1007/s11104-014-2347-7, 2014.

Ries, L. P. and Shugart, H. H.: Nutrient limitations on understory grass productivity and carbon assimilation in an African woodland savanna, Journal of Arid Environments, 72(8), 1423–1430, doi:10.1016/j.jaridenv.2008.02.013, 2008.

Sardans, J. and Peñuelas, J.: Plant-soil interactions in Mediterranean forest and shrublands: impacts of climatic change, Plant and Soil, 365(1–2), 1–33, doi:10.1007/s11104-013-1591-6, 2013.

Sardans, J., Rivas-Ubach, A. and Peñuelas, J.: The C:N:P stoichiometry of organisms and ecosystems in a changing world: A review and perspectives, Perspectives in Plant Ecology, Evolution and Systematics, 14(1), 33–47, doi:10.1016/j.ppees.2011.08.002, 2012.

Steinaker, D. F. and Wilson, S. D.: Phenology of fine roots and leaves in forest and grassland, Journal of Ecology, 96(6), 1222–1229, doi:10.1111/j.1365-2745.2008.01439.x, 2008.

Weiner, T., Gross, A., Moreno, G., Migliavacca, M. and Schrumpf, M.: Following the turnover of soil bioavailable phosphate in Mediterranean Savanna by oxygen stable

isotopes, , doi:10.1029/2017JG004086, 2018.

[Figure]

**Fig. 1.** Schematic diagram of experiment.

**Fig. 2.** Mean diurnal course of NEE (Net ecosystem exchange), LE (Latent heat), H (Sensible heat) and u* (Friction velocity) from March 25th to May 15th 2014.

Open Grassland

Under Canopy

Treatment    CONTROL    N    NP

**Fig. 3.** Total N in the 0-5 cm soil.

OPEN GRASSLAND

UNDER CANOPY

**Fig. 4.** 2M-KCL extractable N in surface soil.

Treatment — CONTROL — N — NP

**Fig. 5.** Olsen-extract phosphate-P in surface soil.

PRETREATMENT

POST-TREATMENT

**Fig. 6.** Leaf N and P contents for pasture plants from 2014 to 2017.

OPEN GRASSLAND

UNDER CANOPY

Treatment  CONTROL  N  NP

**Fig. 7.** Ratio of extractable N (2M KCL) to extractable P (Olsen method) in surface soil.

[Figure]

**Fig. 8.** Herbaceous Layer leaf N:P ratios form 2014 to 2017.

---

## Author Response (AR1)

Dear Editor,

This letter is in reference to our revised manuscript, bg-2018-375.

In most specific cases in the open discussion we agreed with the reviewers and have incorporated the suggested changes into the revised manuscript. For this author response, as we have already provided a point-by-point reply to both sets of reviewer comments during public discussion, we do not repeat these responses in this document.

In the revised manuscript we have altered substantial portions of the text. The major changes, which do not include numerous small text edits, are in summary:

– Substantial rewrites of the introduction and discussion sections of the manuscript

– Justification for the use of a technically non-replicated design based on ecosystem-level eddy covariance footprints, and consideration of the implications of this for the overall interpretation of the experiment

– Addition of additional data regarding pretreatment conditions

– Addition of additional data regarding soil N:P stoichiometry during the period of the study (which also involved adding an additional author)

– Clarification of some of the methods, including correction of ambiguous phrasing and (hopefully) clearer descriptions of the design and sampling protocol

– Addition of a figure illustrating the experimental design and theory behind the experiment

– Corrections to the several mistakes in the figures in the original documents as well as addition of finer scales to timeseries figures

– Removal of some abbreviations for easier readability

We have also added a short supplement showing the N and P concentrations used to assess N:P stoichiometry. We follow with a marked up version of the revised manuscript.

Richard Nair (on behalf of co-authors)

[revised manuscript text omitted]

**Figure 6.** Fine root biomass in top 13 cm from direct soil cores (Nov. 16, Nov. 17) and ingrowth cores (installed Dec.2016 in Mar.17, Nov.17, then installed Nov.17, removed Mar.18, May18).  Root biomass showed a seasonal cycle and also differences between treatments, with more roots in general under canopies and in fertilized plots. Error bars show value ± SE and letters indicate Tukey-HSD groupings for most parsimonious models within treatment for both Open Pasture (OP) and  Tree Covered (TC) locations combined in individual sessions. Both +N and +NP treatments tended to have more root biomass than control treatments. Across the whole dataset, +N had significantly (P < 0.05) more fine roots than the other two treatments.

[Figure]

**Figure 7.** Control Treatment root length density (RLD) response to rain pulse in May  2017. RLD throughout the soil profile (P < 0.001 for both locations) indicating short-term proliferation of root growth.

Comparison of minirhizotron root length density (RLD, mm mm$^{-2}$) dynamics for open pasture with site-level grassland Normalized Difference Vegetation Index (NDVI) and grassland Greenness Colour Coordinate (GCC). After the rain pulse in May 2017 (indicated on graph, light blue), minirhizotron measurements could detect the similar response to above-ground (shown in more detail figure **??**). Desynchronization was evident in both autumn periods where proximal-remote sensed metrics reached near-peak levels while RLD

[Figure]

remained low.

**Figure 8.** Comparison of minirhizotron root length density (RLD, mm mm$^{-2}$) dynamics at 9 cm depth for open pasture with site-level grassland. Normalized Difference Vegetation Index (NDVI) and grassland Greenness Colour Coordinate (GCC). After the rain pulse in May 2017 (indicated on graph, light blue dashes), minirhizotron measurements could detect the similar response to above-ground (shown in more detail figure 7). Desynchronization was evident in both autumn periods where proximal-remote sensed metrics reached near-peak levels while RLD remained low.

**Table 1.** Summary of pretreatment measures within EC footprints ('treatments'). All measurements were made in Nov 2013 for soil and in spring 2014 for plants. All soil properties are measured on surface (0-5 cm) soil and open pasture leaf N:P are a biomass weighted aggregate of samples of grass and forbs ( 98 % of herbacious layer biomass). All errors are standard deviations. Soil nutrient contents (C,N,and Olsen-P) were always significantly higher in UC than OP locations ($P < 0.005$, $P < 0.005$, $P < 0.05$ respectively). There were no significant effects due to footprint except soil pH (*, $p < 0.05$) but this did not result in a difference of weighted mean vegetative N:P ratios, so we considered this unimportant for the overall design.

| Treatment | Control | +N | +NP | Control | +N | +NP |
|---|---|---|---|---|---|---|
| Habitat | Open Pasture (OP) | | | Under Canopy (UC) | | |
| Soil Texture (Clay:Silt:Sand) | 6:20:74 | 5:20:75 | 6:21:73 | 1:20:79 | 5:20:75 | 8:25:67 |
| C (mg g$^{-1}$) | 4.87 ± 1.4 | 5.33 ± 1.2 | 6.56 ± 1.6 | 13.59 ± 0.5 | 10.06 ± 1.2 | 12.57 ± 3.8 |
| N (mg g$^{-1}$) | 0.86 ± 0.1 | 0.93 ± 0.1 | 1.04 ± 0.1 | 1.54 ± 0.1 | 1.30 ± 0.2 | 1.49 ± 0.4 |
| Soil C:N | 5.7 : 1 | 5.7 : 1 | 6.3 : 1 | 8.9 : 1 | 7.8 : 1 | 8.4 : 1 |
| Olsen-Extract P ($\mu$g g$^{-1}$) | 2.3 ± 0.6 | 3.68 ±1.4 | 3.38 ± 2.0 | 5.44 ± 0.9 | 5.45 ± 3.8 | 3.77 ± 1.4 |
| Soil pH | 5.42 ± 0.1 | 5.56 ± 0.5 | 4.93 ± 0.3 * | 5.50 ± 0.6 | 5.58 ± 0.4 | 4.93 ± 0.3 * |

| Treatment | Control | +N | +NP |
|---|---|---|---|
| Herbacious Layer Leaf N:P | 13:1 | 13:1 | 13:1 |

**Table 2.** Absolute (Herbaceous layer) Biomass Measurements and Root:Shoot Ratios at two points in 2017. Root shoot ratio increased later in the growing season but tended to decrease in nutrient added treatments, with the exception of N:OP in March and N:TC in May which exhibited unusually high root:shoot ratio.

**March 2017**

| Nutrient | Vegetation $\mathrm{kg\ ha^{-1}}$ | Aboveground Biomass $\mathrm{kg\ ha^{-1}}$ | Belowground Biomass | Root:Shoot Ratio |
|---|---|---|---|---|
| Control | TC | $290 \pm 20$ | $5770 \pm 320$ | $20 \pm 2 : 1$ |
| Control | OP | $290 \pm 20$ | $4510 \pm 250$ | $15 \pm 1 : 1$ |
| N | TC | $340 \pm 20$ | $6890 \pm 390$ | $21 \pm 2 : 1$ |
| N | OP | $290 \pm 20$ | $7590 \pm 420$ | $26 \pm 3 : 1$ |
| NP | TC | $380 \pm 30$ | $6380 \pm 350$ | $17 \pm 2 : 1$ |
| NP | OP | $300 \pm 30$ | $4900 \pm 270$ | $16 \pm 2 : 1$ |

**May 2017**

| Nutrient | Vegetation | Aboveground Biomass | Belowground Biomass | Root:Shoot Ratio |
|---|---|---|---|---|
| Control | TC | $260 \pm 40$ | $6900 \pm 700$ | $26 \pm 5 : 1$ |
| Control | OP | $370 \pm 60$ | $4280 \pm 330$ | $12 \pm 2 : 1$ |
| N | TC | $220 \pm 30$ | $8250 \pm 640$ | $38 \pm 6 : 1$ |
| N | OP | $400 \pm 50$ | $5960 \pm 580$ | $15 \pm 2 : 1$ |
| NP | TC | $360 \pm 40$ | $7850 \pm 900$ | $22 \pm 3 : 1$ |
| NP | OP | $440 \pm 30$ | $6670 \pm 700$ | $15 \pm 2 : 1$ |

---

## Author Response (AR2)

Dear Editor

This letter is in reference to our revised manuscript, bg-20180375. We are fully in agreement with all of the minor revisions suggested and have made them to the manuscript.

5    – Spelling of 'stoichiometry' corrected in all instances. We also found a repeated incorrect spelling of 'herbaceous' which is also revised.

   – Use of 'borderline' replaced with 'marginal' to reduce colloquial language

   – We agree wholeheartedly with the overuse of abbreviations which were a relic of previous drafts of the manuscript. We have removed all uses of the TC and OP abbreviations, except in figure 1.

10    – Replaced all instances of 'location' when used in reference to the experimental treatment with 'microhabitat'

   – Clarified that root-shoot ratio applies to the herbacious layer only

   – Slightly rewrote the introduction to the discussion.

   – We have also made a few very minor text edits for readability and accuracy which are indicated in the version with marked changes.

15   We hope these edits are satisfactory and thank you for your handling of the revision process.

Richard Nair (on behalf of all co-authors)

[revised manuscript text omitted]